# Interplay between the human gut microbiome and host metabolism

Alessia Visconti [1,9], Caroline I. Le Roy[1,9], Fabio Rosa[1], Niccolò Rossi[1,2], Tiphaine C. Martin[1,3,4], Robert P. Mohney [5], Weizhong Li [6,8], Emanuele de Rinaldis[7], Jordana T. Bell [1], J. Craig Venter[6,8], Karen E. Nelson[6,8], Tim D. Spector[1,10]* & Mario Falchi[1,10]*

The human gut is inhabited by a complex and metabolically active microbial ecosystem. While many studies focused on the effect of individual microbial taxa on human health, their overall metabolic potential has been under-explored. Using whole-metagenome shotgun sequencing data in 1,004 twins, we first observed that unrelated subjects share, on average, almost double the number of metabolic pathways (82%) than species (43%). Then, using 673 blood and 713 faecal metabolites, we found metabolic pathways to be associated with 34% of blood and 95% of faecal metabolites, with over 18,000 significant associations, while species showed less than 3,000 associations. Finally, we estimated that the microbiome was involved in a dialogue between 71% of faecal, and 15% of blood, metabolites. This study underlines the importance of studying the microbial metabolic potential rather than focusing purely on taxonomy to find therapeutic and diagnostic targets, and provides a unique resource describing the interplay between the microbiome and the systemic and faecal metabolic environments.

[1] Department of Twin Research & Genetic Epidemiology, King's College London, London, UK. [2] BioISI—Biosystems & Integrative Sciences Institute, Faculty of Sciences, University of Lisbon, Lisbon, Portugal. [3] Department of Oncological Sciences, Icahn School of Medicine at Mount Sinai, New York, NY, USA. [4] The Tisch Cancer Institute, Icahn School of Medicine at Mount Sinai, New York, NY, USA. [5] Metabolon, Inc., Morrisville, NC, USA. [6] Human Longevity, Inc, San Diego, CA, USA. [7] Immunology & Inflammation, Cluster of Precision Immunology, Sanofi, Cambridge, MA, USA. [8]Present address: J. Craig Venter Institute, La Jolla, CA, USA. [9]These authors contributed equally: Alessia Visconti, Caroline I. Le Roy. [10]These authors jointly supervised this work: Tim D. Spector, Mario Falchi. *email: tim.spector@kcl.ac.uk; mario.falchi@kcl.ac.uk

The human gut is home to trillions of microbes that form a complex community referred to as the gut microbiota. The metabolic activity of the gut microbiota is essential in maintaining host homoeostasis and health, as proven, for instance, by the study of germ-free animals[1,2]. Although the presence of a microbiota is vital, variations in its composition induce metabolic shifts that may result in alterations of host phenotype[3]. The gut microbiome is highly malleable and can be altered throughout lifespan mostly by environmental factors, such as diet and medication[4–6]. Although the external environment plays an important role in shaping the gut microbiome community, the host can affect the microbial ecosystem through its immune system, and has also an impact on the faecal metabolic content[7–9].

The joint study of microbiome and metabolome has been suggested as the most promising approach to evaluate host–microbiome interactions[10]. However, studying the metabolic holobiont is complex, and few studies have tackled this issue in humans at any scale. Our group previously used 16S rRNA gene amplicon data to confirm that the gut microbiome is exceptionally metabolically active, and that the faecal metabolome may improve our estimation of the gut microbiota impact on health[11]. However, it is not possible to fully capture the metabolic activity of the gut microbiome using 16S rRNA gene amplicon-sequencing techniques alone, and the use of the more comprehensive whole metagenomic shotgun sequencing (WMGS) is necessary. Indeed, WMGS not only detects the taxonomic composition at higher resolution but also allows inferring its function, thus allowing the study of the metabolic potential of the microbial community.

Here, we study the effect of this metabolic activity on host health. We assess the impact of the gut microbiome on both the gut and host systemic metabolism by using WMGS and untargeted faecal and blood metabolomics data. We find multiple associations between the gut microbiome (taxonomic composition and microbial metabolic function) and faecal and blood metabolites. In addition, we identify a number of microbial species and metabolic functions likely to play a leading role in the gut-systemic metabolic interplay.

## Results

**Microbial metabolic pathways are shared across subjects**. WMGS was performed on faecal samples provided by 1,054 volunteers from the TwinsUK registry, of which 1,004 survived quality control with an average of 39M microbial reads per sample (see the "Methods" section, Supplementary Table 1). Taxonomic profiling identified, in the kingdoms of archaea and bacteria, 14 phyla, 24 classes, 37 orders, 74 families, 182 genera, and 580 species present in at least one sample (see the "Methods" section). Each species was observed in a median of 2.7% of the samples, and 12% of species were sample-specific (Fig. 1). The most ubiquitous species were from the *Subdoligranulum* genus (unclassified species), *Ruminococcus obeum*, *Ruminococcus torques*, and *Faecalibacterium prausnitzii*, all detected in more than 98% of the samples (Supplementary Fig. 1). Microbial metabolic detection (as described by the MetaCyc microbial metabolic pathways) identified 434 non-redundant pathways, which were detected in most samples (see the "Methods" section). Each pathway was observed in a median of 91.6% of the samples, with 12% of the pathways present in all samples and only 2% being sample-specific (Fig. 1).

Microbial metabolic pathways were widely shared between individuals, compared to their taxonomical composition. Indeed, multiple known species (up to 465, and 29 on average) identified from the WMGS data, plus a large number of unclassified species,

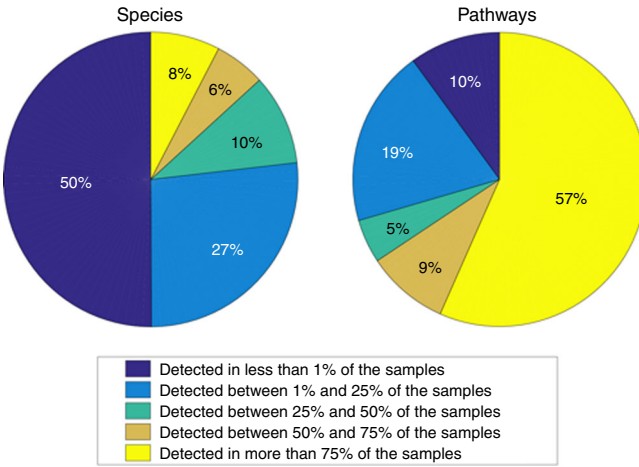

**Fig. 1** Gut microbiome composition. The composition of the gut ecosystem is unique to an individual while its functionality is maintained across the population. Pie charts represent the percentage of species (on the left) and microbial metabolic pathways (on the right) present in <1% of the population (dark blue), between 1% and 25% (light blue), between 25% and 50% (turquoise), between 50% and 75% (brown), and more than 75% (yellow)

contributed to the abundance of each microbial metabolic pathway (Supplementary Data 1). As a consequence, pathway prevalence within our sample strongly correlated with the number of species in which it could be detected (Spearman's $\rho = 0.34$; $P = 9.4 \times 10^{-9}$), i.e., pathways present in the largest number of species were also those with highest prevalence (and vice versa). When comparing pairs of unrelated individuals, we observed that, on average, they shared 82% of the pathways but only 43% of the species (paired Wilcoxon's test $P < 2 \times 10^{-16}$, Supplementary Fig. 2, see the "Methods" section).

**Microbiome and faecal metabolic content are strongly linked**. Faecal metabolomics and WMGS data were available for 479 individuals, and generated on the same faecal samples. 713 annotated metabolites were measured in more than 50 individuals and tested for association with the gut microbiome at both taxonomic and functional levels using PopPAnTe[12], which uses a variance component framework and the matrix of the expected kinship between each pair of individuals to model the resemblance between family members. Sex and age at sample collection were included as covariates (see the "Methods" section, Fig. 2). As expected, both the composition of the gut microbiome and its metabolic function were widely associated with the faecal metabolic content. At a 5% false discovery rate (FDR) we found 16,133 associations with microbial metabolic pathways and 2493 associations with microbial species (Supplementary Data 2 and 3). In particular, 99.7% of the metabolic pathways were significantly associated with 95% of the faecal metabolites, while 90% of the species were associated with 82% of the faecal metabolites (see the "Methods" section; Fig. 2). We observed 48% and 51% positive associations with microbial metabolic pathways and species, respectively. On average, each metabolite level was associated with 4 species and 24 pathways. In addition, 145 (20%) metabolites were associated to a single species, while only 50 of them (7%) were associated to a single pathway. Five microbial species played a major metabolic role and were independently associated with 10% of the faecal metabolites (Supplementary Fig. 3): unclassified *Subdoligranulum* spp. (149 metabolites), *Akkermansia muciniphila* (106 metabolites), *Roseburia inulinivorans* (105 metabolites), *Methanobrevibacter smithii* (96 metabolites), and

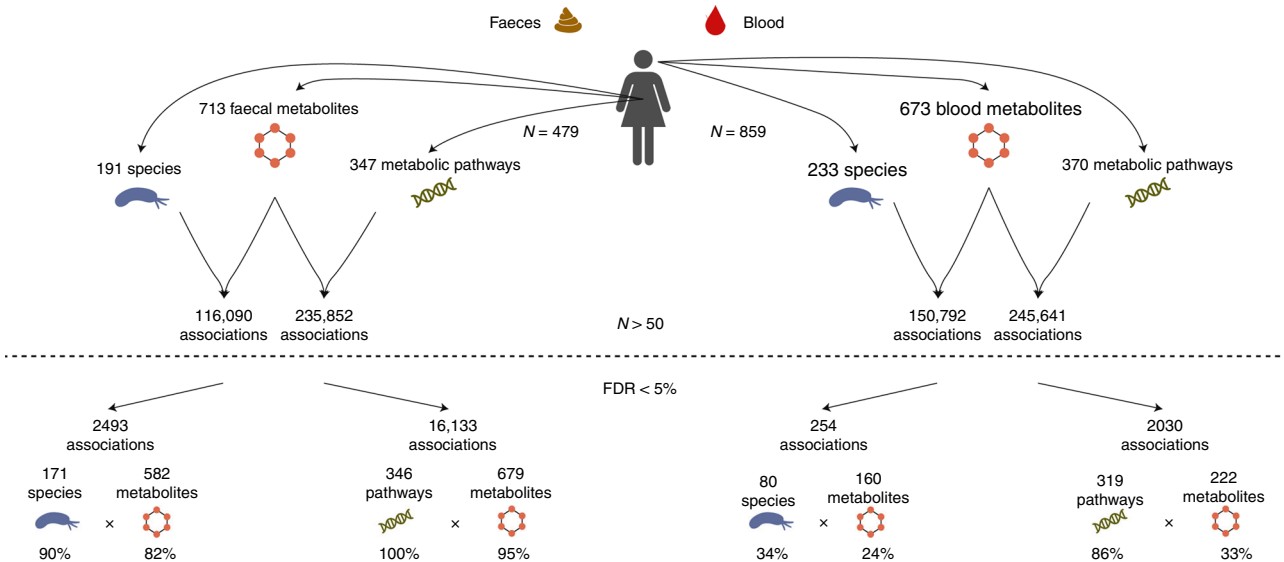

**Fig. 2** Study design and number of associations. The top of the figure reports the number of microbial species and metabolic pathways which were detected in at least 50 individuals with metabolomics and WMGS data, and that were used in the study, and the number of associations tested. The bottom of the figure reports the number of associations that were significant at a 5% FDR, along with the number and percentage of metabolites, microbial species, and microbial metabolic pathways involved. Association testing was performed using PopPAnTe[12], in order to model the resemblance between family members. Sex and age at the sample collection were included as covariates

*Roseburia intestinalis* (92 metabolites). In contrast, the top-five microbial metabolic pathways were associated with more than 53% of the faecal metabolites, with the pathways of L-rhamnose degradation I, Kdo transfer to lipid IVA III (Chlamydia), CDP diacylglycerol biosynthesis I and II and NAD biosynthesis I from aspartate associating with 226, 218, 215, 215, and 206 faecal metabolites, respectively.

We calculated the enrichment of the associated metabolites for metabolic super-pathways (as annotated by Metabolon, Inc.; see the "Methods" section). Faecal metabolites associated with microbial species were enriched for a decrease in amino acids (PAGE adj $P = 1.6 \times 10^{-4}$) and an increase in lipids (PAGE adj $P = 1.9 \times 10^{-3}$), while metabolites associated with metabolic pathways were enriched for a decrease in lipids (PAGE adj $P = 8.0 \times 10^{-5}$), and an increase in both nucleotides (PAGE adj $P = 0.02$) and carbohydrates (PAGE adj $P = 0.03$).

B vitamins in faeces were strongly associated with both species and metabolic pathways, with riboflavin (vitamin B2), nicotinate (vitamin B3), pantothenate (vitamin B5), pyridoxine (vitamin B6), biotin (vitamin B7) associated with 9–27 species and with 48–155 microbial pathways (Supplementary Data 2 and 3). Finally, 16 associations were observed between faecal vitamin E (alpha, beta, gamma and delta tocopherol) and species/pathways.

Eleven of the 82 drug or drug-derived metabolites detected by the Metabolon platform in faeces were present in at least 50 samples with matching metagenomics data. At the species level, we observed six associations with three of these metabolites passing an FDR threshold of 5% (Supplementary Data 2). One association was between 3-hydroxyquinine (a degradation product of quinine, used against malaria but also contained as a flavouring in beverages, including tonic water) and unclassified *Anaerotruncus* spp. ($\beta = 0.68$, SE = 0.18. $P = 4.02 \times 10^{-5}$). Two negative associations were identified between salicylic acid (a precursor of aspirin) and *M. smithii* ($\beta = -0.53$, SE = 0.17, $P = 2.21 \times 10^{-4}$) and unclassified *Anaerotruncus* spp. ($\beta = -0.62$, SE = 0.17, $P = 2.13 \times 10^{-4}$). Finally, N-carbamylglutamate (a drug that can be used for the treatment of hyperammonemia) was associated with *F. prausnitzii* ($\beta = 0.68$, SE = 0.18, $P = 2.21 \times 10^{-4}$), *Odoribacter splanchnicus* ($\beta = 0.88$, SE = 0.25, $P = 3.93 \times 10^{-4}$), and *Blautia*

*hydrogenotrophica* ($\beta = -0.57$, SE = 0.15, $P = 1.17 \times 10^{-4}$). At a 5% FDR, a total of 101 associations were observed between microbial metabolic pathways and faecal metabolites annotated as drugs or and drug-derived metabolites (Supplementary Data 3). Namely: 3-(N-acetyl-L-cystein-S-yl) acetaminophen (26 associations, metabolite derived from paracetamol), 3-hydroxyquinine (1 association), 4-acetamidophenol (24 associations, metabolite derived from paracetamol), carboxyibuprofen (2 associations, metabolite derived from ibuprofen), N-carbamylglutamate (8 associations) and salicylic acid (40 associations).

**The microbiome associates with host systemic metabolites.** Blood metabolomics profiling was available for 859 individuals with WMGS data. Faecal and blood samples were collected, on average, 0.9 years apart, with 41% of our samples collected within one week, and 91% within 2 years (Supplementary Fig. 4). Intra-individual correlation analysis of the tested metabolites showed a good correlation between samples collected up to 2 years apart ($n = 149$, mean Pearson's $\rho = 0.53$, SD: 0.12, 1st–3rd interquartile range: 0.47–0.60), as confirmed by permutation analysis ($P_{empirical} = 1 \times 10^{-4}$, see the "Methods" section). We further observed that metabolomics stability persists over longer periods of time (Supplementary Table 2), in line with previous literature suggesting that human metabolic profiles are conserved for up to 7 years[13].

Six hundred and seventy-three annotated metabolites (including 369 metabolites also measured in faeces) were measured in more than 50 individuals, and used in this study. At a 5% FDR, we identified 2030 associations with microbial metabolic pathways and 254 associations with microbial species, of which 44% and 43% were positive, respectively (Fig. 2; Supplementary Data 4 and 5). In particular, 86% and 34% of the microbial metabolic pathways and species associated with 33% and 24% of the studied blood metabolites, respectively, with a total of 309 unique blood metabolites (46%) associated with the microbiome. The species showing the largest number of associations with blood metabolites were *Lactobacillus acidophilus* ($n = 30$), and *Lactobacillus fermentum* ($n = 14$; Supplementary Fig. 5). The metabolite

**Table 1 Association between microbial species and the same named metabolite in both faeces and blood**

| Species | Metabolite | $F_N$ | $F_\beta$ | $F_{SE}$ | $F_P$ | $B_N$ | $B_\beta$ | $B_{SE}$ | $B_P$ |
|---|---|---|---|---|---|---|---|---|---|
| *Akkermansia muciniphila* | *p*-cresol sulfate | 443 | −0.35 | 0.11 | $7.74 \times 10^{-4}$ | 829 | 0.47 | 0.10 | $6.36 \times 10^{-6}$ |
| *Bacteroidales bacterium ph8* | Sebacate | 406 | 0.68 | 0.18 | $1.65 \times 10^{-4}$ | 718 | −0.50 | 0.09 | $4.33 \times 10^{-9}$ |
| *Eubacterium rectale* | *p*-cresol sulfate | 444 | 0.45 | 0.11 | $2.84 \times 10^{-5}$ | 823 | −0.45 | 0.11 | $2.09 \times 10^{-5}$ |
| *Methanobrevibacter smithii* | Threonate | 287 | −0.65 | 0.14 | $3.31 \times 10^{-6}$ | 527 | 1.01 | 0.16 | $4.09 \times 10^{-10}$ |
| *Oscillibacter* spp. | 3-phenylpropionate | 418 | −0.54 | 0.11 | $1.95 \times 10^{-6}$ | 799 | −0.55 | 0.10 | $1.02 \times 10^{-7}$ |
| *Roseburia inulinivorans* | *p*-cresol sulfate | 433 | 0.43 | 0.10 | $1.09 \times 10^{-5}$ | 802 | −0.53 | 0.10 | $1.45 \times 10^{-7}$ |
| *Subdoligranulum* spp. | *p*-cresol sulfate | 461 | −0.32 | 0.10 | $9.86 \times 10^{-4}$ | 854 | 0.62 | 0.10 | $1.75 \times 10^{-9}$ |

For each association, in faeces (F) and blood (B), we report the number of observations (N), effect size ($\beta$), standard error (SE), and P value (P)

sebacate showed the highest number of associated species ($n = 11$), followed by tartronate ($n = 9$), phenylacetylglutamine ($n = 8$), and *p*-cresol sulfate ($n = 6$). On average, each blood metabolite was associated with two species (with 118 species associating with a single metabolite) and 10 metabolic pathways (with 93 pathways associating with a single metabolite). The three microbial metabolic pathways showing the largest number of associations with blood metabolites were the super pathways of L-phenylalanine and of L-alanine biosynthesis, and the pathway of urate biosynthesis/inosine 5'-phosphate degradation (30, 26, and 24, respectively). Four blood metabolites were associated with more than 100 microbial metabolic pathways: phenylacetylglutamine ($n = 143$) and *p*-cresol-glucuronide ($n = 102$), two known gut microbial-derived metabolites, as well as tyramine O-sulfate ($n = 130$), that can be synthesised by a *Eubacterium* enzyme[14], and 1,5-anhydroglucitol ($n = 129$), that is present in a wide variety of food products.

**Pathways have a broader metabolic footprint than species**. Overall, we identified about seven times more associations between faecal and blood metabolites and microbial metabolic pathways than microbial species. We observed that pathways found in a larger number of species have a stronger impact on the metabolome, with a significant positive correlation between the number of species contributing to each pathway and the number of associations between the pathway and both faecal and blood metabolites (Spearman's $\rho = 0.27$, $P = 2 \times 10^{-6}$, and $\rho = 0.33$, $P = 1 \times 10^{-9}$, respectively).

Our results confirmed a wide network of associations between the gut microbiome and the faecal metabolome, which extends to the systemic metabolome. At a 5% FDR, we identified 360 microbial metabolic pathways associating with 679 faecal and 222 blood metabolites, and 233 microbial species associating with 582 faecal and 160 blood metabolites. We observed that age at the sample collection had a negligible effect on the number of significant associations identified between faecal and blood metabolites and both microbial metabolic pathways and species (see the "Methods" section, Supplementary Table 3, Supplementary Data 6–9, Supplementary Fig. 6). Similarly, correction for drug intake (antibiotics, metformin, and proton–proton inhibitor (PPI)), which was assessed in a small subset of our study sample ($n = 411$, see the "Methods" section), appeared to minimally affect the number of significant associations between the metagenome and the faecal and blood metabolome (Supplementary Table 4, Supplementary Data 10–13).

Notably, in both faeces and blood, the majority of the metabolic pathways were associated with metabolites apparently unrelated to their functions. Indeed, only 999 out of 4,891 unique faecal metabolite–pathway associations (20%) and 186 out of 419 unique blood metabolite–pathway associations (44%), respectively, linked 155 faecal and 42 blood metabolites to pathways either producing or consuming them (see the "Methods" section).

Most microbial metabolic pathways (85%) associating with one or more metabolites in faeces also associated with one or more metabolites in blood (Supplementary Table 5). In contrast, the majority of microbial species only associated with faecal metabolites alone (58%, Supplementary Table 5). Still, 31% of species showed association with both faecal and blood metabolites, suggesting important effects on host systemic metabolism for this subset. Specifically, 4,861 pairs of faecal–blood metabolites co-associated with the same species and 108,565 pairs with the same metabolic pathway. Among these associations, 152 pairs involved exactly the same named metabolite in both faeces and blood (145 with metabolic pathways, and only seven with species; Table 1, Supplementary Data 14), while 113,274 pairs involved a different metabolite in faeces and blood (unique pairs $n = 27,608$). Sebacate, threonate, and *p*-cresol sulfate, in both faeces and blood, showed the largest number of associations with pathways and with species in both faeces and blood.

**The microbiome interfaces faecal and systemic metabolism**. We further investigated the full set of faecal–blood co-associating pairs of metabolites to better understand whether the observed associations were randomly coincident at the same species or pathway, or if they were suggesting an interplay between the gut and systemic environments. We assessed, through simulations, the probability that the microbiota was involved in the dialogue between faecal and blood metabolites (see the "Methods" section). We hypothesised that, if the species (or metabolic pathway) was involved in the dialogue between faecal and blood metabolites, these were expected to more strongly correlate in individuals for which the species (or metabolic pathway) was present than in the remaining samples. Significantly higher correlations were observed between co-associated faecal–blood metabolite pairs when species (or metabolic pathways) were detected ($P_{\text{empirical}} = 1 \times 10^{-3}$ for species, and $P_{\text{empirical}} = 0.03$ for pathways). These results suggested that at least some of the observed faecal and blood metabolite associations were likely not randomly coincident at the same species (or pathway), thus supporting the analysis of this subset of faecal metabolite–blood metabolite–species/pathway trios with the P-gain approach[15] (Supplementary Fig. 7). The P-gain statistic compares the increase in strength of association with the species (or pathway) when using the metabolite ratios compared to the smaller of the two P values when using the two metabolite abundances individually. A strong reduction in P value indicates that two metabolite levels may be linked by a mechanism that involves the gut microbiota. To carefully assess a significance threshold for the P-gain statistics in our sample, we estimated its empirical null distribution through simulations. We obtained a P-gain threshold of 73 for species and of 42 for metabolic pathways at an experimental-wide $\alpha$-level of 0.05 (see the "Methods" section). P-gains passing these thresholds are reported in Supplementary Data 15 and 16, and included 31% of the P-gains with species

(1325/4232 co-associated metabolite pairs) and 19% with microbial metabolic pathways (16,839/88,452 co-associated metabolite pairs). The $P$-gain statistics suggested a potential dialogue between 36% of the faecal metabolites and 5% of the blood metabolites, involving 12% of the species ($n = 29$). This dialogue was wider with the microbial metabolic pathways, involving 70% of the faecal and 14% of blood metabolites, and 67% of the pathways ($n = 247$).

At the species level, unclassified *Subdoligranulum* spp. accounted for 49% of the putative dialogue, and *F. prausnitzii*, *R. inulinivorans*, *M. smithii*, *E. rectale*, and *A. muciniphila* together contributed to a further 36%. In contrast, the results at the pathway level were not dominated by a limited number of pathways, with the top six contributing only towards 24% of the observed dialogue.

**Methanogens associate with adiposity**. Threonate in blood showed the highest $P$-gains involving a large number of significantly co-associated faecal metabolites ($n = 61$ faecal metabolites, including threonate levels in faeces). All these associations involved the methanogenic *M. smithii*, the main archeon in the human gut[16], present in 62% of our metagenomic samples. Threonate is produced from vitamin C under oxidative conditions[17]. In both blood and faeces, threonate was also associated with two microbial pathways linked to methanogenesis: coenzyme factor 420 biosynthesis ($\beta = 0.94$, SE $= 0.18$, $P = 2.2 \times 10^{-7}$) and methanogenesis from $H_2$ and $CO_2$ ($\beta = 0.93$, SE $= 0.18$, $P = 3.3 \times 10^{-7}$), to which *M. smithii* contributes, in our sample, for about 47% (the remaining attributable to *Methanosphaera stadtmanae*, <1%, and to unclassified species, 53%; see the "Methods" section). The role of *M. smithii*, and of other methanogenic microbes in human health is still unclear, however, several studies suggested that its depletion is linked to obesity[18,19]. We found it to be significantly negatively associated with the percentage of visceral fat ($\beta = -0.09$, SE $= 0.04$, $P = 0.013$; Supplementary Table 6). We also observed a significant negative association ($P < 0.05/3 = 0.017$, Supplementary Table 6) between blood threonate and three measures of adiposity, namely BMI ($\beta = -0.48$, SE $= 0.12$, $P = 3.2 \times 10^{-5}$), and the percentages of total body fat ($\beta = -0.41$, SE $= 0.10$, $P = 4.3 \times 10^{-5}$) and visceral fat ($\beta = -0.48$, SE $= 0.11$, $P = 2.6 \times 10^{-5}$), while faecal threonate was not associated with any measure of adiposity ($P > 0.05$). On the other hand, 31 out of 61 faecal metabolites whose dialogue with blood threonate via *M. smithii* was confirmed by the $P$-gain statistic were significantly associated with measures of adiposity ($P < 0.05/(61 \times 3) = 1.3 \times 10^{-3}$; Supplementary Data 17).

### Discussion

Microbiome studies are mainly focused on the effect of individual microbial taxa on human health, while the metabolic potential of microbes has been largely overlooked.

A previous report on a small sample of female subjects ($n = 18$) showed that, despite a high $\beta$-diversity at the phyla level, between 26% and 53% of the 'enzyme'-level functional groups were shared among samples[20]. Higher similarity of microbial metabolic pathways vs. organismal abundances was also observed by the larger ($n = 242$) Human Microbiome Project[21]. This may be explained by the high redundancy of metabolic pathways across different microbial species[22]. Our larger study validates these findings, and estimates that 12% of the microbiome metabolic potential (as described by the MetaCyc microbial metabolic pathways) is present in all individuals. More in general, a random pair of unrelated subjects shares on average 82% of their microbial metabolic pathways, while this is the case for only 43% of the

species. We also observed that microbial metabolic pathways are highly redundant, with up to 465 identified species (and a possibly large number of unknown ones) sharing the same metabolic pathway.

Using 713 faecal and 673 blood metabolites measured by Metabolon, Inc. and WGMS data, we conducted a microbiota-wide association study. Our results showed that the gut metagenome (both at the species and at the metabolic pathway levels) widely associates with both the gut and host systemic metabolism. At a 5% FDR, we identified association between the faecal metabolites and 90% of the microbial species and 99.7% of the microbial metabolic pathways. In particular, metabolic pathways were significantly associated with 95% of the faecal metabolites, while microbial species were associated with 82% of the faecal metabolites. The results at the taxonomic level were comparable to those previously reported in a recent study on the TwinsUK cohort leveraging 16S rRNA gene amplicon data[11]. In both studies, we observed that over 90% of microbes were associated with a vast proportion of the measured gut metabolites (>80%). The WMGS data used in this study allowed us to extend these observations, by improving the precision of the taxonomic associations at the species level rather that at the genus level. For instance, we were able to identify five species interacting with at least 10% of the studied faecal metabolites. Four of them (*Subdoligranulum* spp., *A. muciniphila*, *R. inulinivorans*, and *R. intestinalis*) were present in at least 80% of the population sample and are already known for their ability to affect faecal metabolic content[23–26]. Additionally, the WMGS data allowed the inference of microbial metabolic pathways and their association with the faecal metabolome, which could not be performed on the previous TwinsUK study.

Interestingly, among the numerous microbiome–metabolome associations identified in this study, a large proportion was involved with the metabolism of vitamins. For instance, we observed over 700 associations with vitamin B-related metabolites. While B vitamins are mostly provided to the host through diet, these can also be synthesised by lactic acid bacteria[27]. Our results show a similar number of positive and negative associations with vitamin B metabolites, suggesting that the microbiome is not only involved in the biosynthesis of vitamins B but also in its degradation. Drugs can be metabolised by the gut microbiota, and they may affect both the metabolic activity of the gut microbiome and its composition[28,29]. In our analyses, we identified associations between six species and 101 microbial metabolic pathways and 6 out of 11 drugs and drug-related metabolites detected in faeces through the Metabolon platform in a sufficient number of subjects.

In this study, we also evaluated the impact of the gut microbiome on the host systemic metabolism. We showed that nearly half of the blood metabolites ($n = 309$, 46%) were associated with microbial species and/or metabolic pathways. More exactly, 34% of the species and 86% of the pathways were associated with 24% and 33% of the metabolites, respectively. Two bacteria stood out as playing a major role: *L. acidophilus* (5% of the associations) and *L. fermentum* (2% of the associations), both known for their probiotic properties[30–32]. Notably, a previous study on the TwinsUK cohort observed that 72% of blood metabolites were under host genetic influence[33]. Interestingly, 144 out of 309 microbiome-associated blood metabolites (47%) identified in our study were not heritable. Heritabilities for the remaining 165 blood metabolites ranged from 10% to 78%, with a mean value of 47% (Supplementary Data 18). This suggests that, despite the widespread host genetic effects on blood metabolites, the gut microbiome might play a role on the systemic metabolism that is independent from the host genome.

In our sample, composed predominantly of active middle-aged women, we observed that associations between the gut microbiome and both the gut and systemic metabolisms were minimally impacted by age. This is in line with previous observations showing that, in absence of external perturbation, the gut microbiome of healthy adults remains relatively stable for years[34,35]. We also observed, in a small subset of our study sample, that antibiotics, metformin, or PPI intake had a minimal effect on the associations between the microbiome and the metabolome, although this is likely due to the limited number of individuals taking any of these three drugs.

Bile acids (BAs) metabolism has been associated with gut microbiota composition in many studies. Indeed, the gut microbiota shapes the composition of the BA pool (by hydrolysis and hydroxy group dehydrogenation of primary BAs to secondary BAs) and BAs can affect the growth of certain gut bacteria[36–38]. In faeces, 5% and 3% of the total number of associations between faecal metabolites and metabolic pathways and species, respectively, were with BAs, over 80% of which were with secondary BAs. In blood, 6% of all associations with species and 3% of all associations with metabolic pathways were with BAs. Again, secondary BAs were more associated (over 70% of all BAs associations) than primary BAs with both species and metabolic pathways.

Sebacate was the faecal metabolite that was associated with the largest number of species and metabolic pathways. Sebacate metabolism has been poorly studied. However, a pharmacokinetic study of sebacate in rats has revealed, post-ingestion, a low systemic bioavailability, suggesting that this may be explained by direct beta-oxidation of sebacate (i.e., sebacate degradation) by the liver, and that only traces of the compound could be detected in faeces[39]. Another study on rats also revealed the absence of sebacate in faeces after intravenous injection of the radioactive compound[40], indicating that it is unlikely that systemic sebacate level affects the gut microbiome through its excretion in the gut. Sebacate was also used as primary carbon source by some gut commensals (Pseudomonas aeruginosa and Pseudomonas multilivoran)[41]. Thus, the observed low post-ingestion level of sebacate in both faeces and blood in rats, and the numerous associations identified by our study between faecal and blood sebacate and the gut microbiome may also be due to its utilisation by gut bacteria as carbon source. Endogenous sebacate, naturally found in blood, can be synthesized, in rats, through omega-oxidation in starvation periods, before undergoing beta-oxidation to produce succinate and be used as energy source through gluconeogenesis[42,43]. It was also reported that gut bacteria may affect liver beta-oxidation through modulation of the immune system in mice[44]. Therefore, an alternative/complementary hypothesis might be that the high number of associations observed between blood sebacate and the gut microbiome might picture the effect that the gut microbiome exerts on liver functions[45].

Altogether, our results indicate an intense interplay between the gut microbiome and its host. While only a small number of metabolites were found to be associated to the same species (or pathway) in both metabolic environments ($n = 152$), we detected more than 27,000 unique pairs of faecal–blood metabolites, which were associated with the same microbial species and/or metabolic pathway (co-associated metabolites). The limited size of our study sample makes it unsuitable to test causality using a Mendelian randomisation method[46]. Nonetheless, using two complementary approaches, we showed that, first, co-associated metabolites are more strongly correlated in the presence of the associated species or metabolic pathways ($P_{empirical} = 1 \times 10^{-3}$ and 0.03, respectively), and, second, that a significant dialogue, as assessed through the $P$-gain statistic, exists between 71% of the faecal and the 15% of blood metabolites, involving 12% of the species

and 67% of the pathways. We highlight four potential mechanisms that could underlie the interplay between these two metabolomic environments (Fig. 3). First, the interplay could be triggered by the metabolic activity of the microbiome[47]. Second, the gut microbiome could mediate metabolite transfer through the gut barrier by affecting its integrity, as suggested, for example, by the associations involving the same species and named metabolites in both blood and faeces (Table 1). Indeed, these associations showed opposite direction of effects, suggesting that microbes may modulate the absorption of the metabolites by the host rather than its bioavailability. Third, microbial growth could be impacted by secretion of metabolites by the host within the gut as extensively discussed regarding BAs[37,48]. Fourth, the host–gut microbiome interplay could also be triggered by non-metabolic interactions including microbial secretion of peptides or direct cell–cell interactions[49], which could not be investigated in the present study.

We observed about seven times more associations between metabolites and microbial metabolic pathways than species. This trend was even stronger when studying the faecal–blood dialogue, with nearly 13 times more co-associated metabolite pairs identified by means of the $P$-gain statistics for microbial metabolic pathways than species. These results support the claim that looking at functions rather than taxonomy alone gives a better appreciation of the true gut microbiome metabolic activity[10]. We suggest that this large number of associations with metabolic pathways is likely due to functional redundancy. Nonetheless, the majority of the metabolic pathways, especially in faeces, were associated with metabolites apparently unrelated to their functions, with only 20% and 44% of the faecal metabolite–pathway associations and blood metabolite–pathway associations linking metabolites with the MetaCyc metabolic pathways either producing or consuming them. Therefore, we cannot exclude that part of the observed associations with pathways are driven by the concerted action of microbial sub-communities rather than only by the specific function of the pathways.

This study has some limitations. First, we used data from a cohort including only individuals of European ancestry and composed predominantly of middle-aged woman (96%, average age 65 years old). Therefore, our results may not generalise to diverse populations. Ideally, data collected in other larger cohorts and meta-analyses would be necessary to confirm our novel findings. Second, despite the large-scale sample, this is a cross-sectional study, and no causal relationship between the microbiome and the metabolome can be inferred from the identified associations. Third, while WMGS data allow us to infer the functional capability of the microbial community, it does not provide information on which microbial metabolic pathways are actually active. Metatranscriptomic data will help in bridging this gap, also allowing discerning between associations with microbial metabolic pathways that are connected to their specific function or that are simply a proxy for microbial sub-communities. Fourth, stool consistency and microbial cell count, which can have an influence on the gut microbiota composition[50,51], were not recorded in this study. Finally, the results obtained in this study are not quantitative, since all analyses were carried out using relative abundances. This implies that the identified associations report the effect of microbial species/metabolic pathways proportion rather than of their actual concentration.

In conclusion, we first confirmed the key role played by the microbiome on the faecal and host systemic metabolism. Next, we described the microbiome effect on the interplay between the two metabolic compartments. We observed that only a few key species, but many common microbial functions, are substantially associated with faecal and blood metabolic profiles. Therefore, microbial metabolic pathways should be considered beyond their

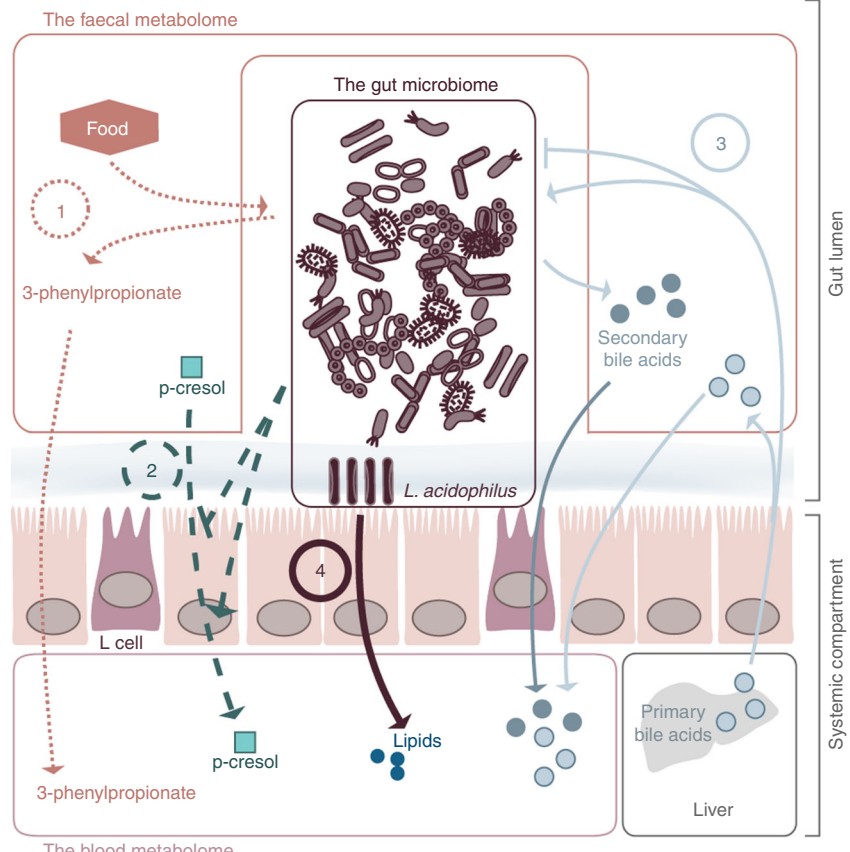

**Fig. 3** Potential mechanisms implicated in the interplay between the gut microbiome, the faecal metabolome, and the blood metabolome. (1) Small dashed lines: metabolites are produced by the microbiota and then absorbed, resulting in associations between the microbiome and both the blood and faecal metabolites. (2) Large dashed lines: the microbiome affects the gut barrier integrity, resulting in alterations of metabolites absorption (i.e., the same metabolite is associated with a species/pathway in both blood and faeces, but the directions of effects are opposite). (3) Light continuous line: metabolites produced by the host, such as bile acids, affect microbial growth. (4) Bold continuous line: direct microbiome to host cell interactions that result in host systemic modulation (i.e., species are associated with blood metabolites but not with faecal metabolites)

primary function and interpreted as proxies for microbial communities, interacting with their surrounding environment. Future treatments designed to improve host health through the modulation of the gut microbiome should optimally target functionally related microbial communities rather than single organisms. Moreover, with this study, we make available to the scientific community a unique resource providing a detailed investigation of the dialogue between the microbiome and the faecal and blood metabolome, which will help in pinpointing potential biomarkers and targets capable of modulating the abundances of metabolites and of species and functions relevant for human health for further investigations, and inform microbiology research on potential new metabolic functions of the gut microbiome. Results are made fully available through extensive Supplementary Materials and through a Web portal (http://www.metabgut.org) where they can be queried and visualised both graphically and as interactive tables.

## Methods

**TwinsUK cohort.** The TwinsUK adult twin registry includes about 14,000 subjects, predominantly females, with disease and lifestyle characteristic similar to the general UK population[52]. Metagenomics sequencing was performed on 1,054 randomly selected samples, while faecal and blood metabolomics was assessed in 479 and 859 individuals with metagenomics data, respectively.

Twins collected faecal samples at home, and the samples were refrigerated for up to 2 days prior to their annual clinical visit at King's College London, when they were stored at −80 °C for an average of 2.3 ± 1.0 years at −80 °C before processing.

Both faecal metabolomics and WMGS data were generated on the same faecal samples. Blood samples, collected during the clinical visit, were stored at −80 °C for an average of 1.8 ± 1.2 years before processing. Faecal and blood samples were collected, on average, 0.9 ± 1.3 years apart.

St. Thomas' Hospital Research Ethics Committee approved the study, and all twins provided informed written consent.

**DNA extraction, library preparation, and sequencing.** A 3-mL volume of lysis buffer (20 mM Tris–HCl pH 8.0, 2 mM sodium EDTA 1.2% Triton X-100) was added to 0.5 g of stool sample, and the sample vortexed until homogenised. A 1.2 mL volume of homogenised sample and 15 mL of Proteinase K (Sigma Aldrich, PN.P2308) enzyme was aliquoted to a 1.5 mL tube with garnet beads (Mo Bio PN. 12830-50-BT). Bead tubes were then incubated at 65 °C for 10 min and then 95 °C for 15 min Tubes were then placed in a Vortex Genie 2 to perform bead beating for 15 min and the sample subsequently spun in an Eppendorf Centrifuge 5424. 800 μL of supernatant was then transferred to a deep well block and DNA extracted and purified using a Chemagic MSM I (Perkin Elmer) following the manufacturer's protocol. Zymo Onestep Inhibitor Removal kit was then performed following manufacturer's instructions (Zymo Research PN. D6035). DNA samples were then quantified using Quant-iT on an Eppendorf AF2200 plate reader.

Nextera XT libraries were prepared manually following the manufacturer's protocol (Illumina, PN. 15031942). Briefly, samples were normalised to 0.2 ng ml$^{-1}$ DNA material per library using a Quant-iT picogreen assay system (Life Technologies, PN. Q33120) on an AF2200 plate reader (Eppendorf), then fragmented and tagged via tagmentation. Amplification was performed by Veriti 96-well PCR (Applied Biosystems) followed by AMPure XP bead cleanup (Beckman Coulter, PN. A63880). Fragment size for all libraries were measured using a Labchip GX Touch HiSens. Sequencing was performed on an Illumina HiSeq 2500 using SBS kit V4 Chemistry, with a read length of 2 × 125 bp. Sequencing of 1054 samples yielded an average number of reads of 54M per sample before quality control.

**Taxonomic profiling and functional annotation**. Paired-end reads were processed using the YAMP pipeline (v. 0.9.1)[53]. Briefly, we first removed identical reads, potentially generated by PCR amplification[54]. Next, reads were filtered to remove adapters, known artefacts, phix174, and then quality trimmed (PhRED quality score < 10). Reads that became too short after trimming ($N < 60$ bp) were discarded. We retained singleton reads (i.e., reads whose mate has been discarded) in order to retain as much information as possible. Contaminant reads belonging to the host genome were removed (build: GRCh37). Low-quality samples, i.e., samples with <15M reads after QC were discarded ($n = 4$). Next, MetaPhlAn2[55] (v. 2.6.0) and the HUMAnN2 pipeline[56] (v 0.10.0), both included into the YAMP pipeline, were used to characterise the microbial community composition and its functional capabilities, respectively. Functional capabilities of the microbial community were described by the MetaCyc metabolic pathways, and assessed using the UniRef90 proteomic database annotations. HUMAnN2 was also used to evaluate the percentage of species contributing to the abundance of each microbial metabolic pathway.

A principal component analysis evaluated using the taxonomic profiling was used to identify and discard ecologically abnormal samples ($n = 37$). If sample scores were greater than three times the standard deviation on one of the first 10 principal components the sample was labelled as outlier and discarded. Finally, we removed individuals not of European ancestry ($n = 9$, self-reported via questionnaire) resulting in 1004 samples with an average number of reads of 39M (39 males, 965 female), all living in the UK at the time of specimen collection (Supplementary Table 1). The dataset included 161 monozygotic twin pairs ($n = 322$), 201 dizygotic twin pairs ($n = 402$), and 280 singletons.

Taxonomic and microbial metabolic pathways relative abundances were arcsine square-root transformed, filtered for outliers using the Grubbs outlier test (significance threshold $P = 0.05$), and standardised to have zero mean and unit variance[57]. Under the assumption that a zero relative abundance meant impossibility to detect the taxum/pathway rather than its absence, zero values were considered as not available (NA).

**Metabolomics profiling**. Metabolite ratios were measured from faecal samples and blood by Metabolon, Inc., Morrisville, NC, USA, by using an untargeted UPLC–MS/MS platform. Details to help reproducing the present findings using comparable non-commercial methodologies are available in the Supplementary Methods and in Zierer et al.[11], for faecal metabolome, and Long et al.[33], for blood metabolome. Briefly, faecal samples were lyophilised then extracted at a constant per-mass basis while blood samples were used directly for extraction at a constant per-volume basis. Proteins and other macromolecules were removed using methanol precipitation. Samples were run using four different methods, against three controls (a pooled sample, extracted water -blank- and a cocktail of standards). Metabolites were identified by comparison to a referenced library of chemical standards[58], and area-under-the-curve analysis was performed for peak quantification and normalised to day median value. To ensure high quality of the dataset, control and curation processes were subsequently used to ensure true chemical assignment and remove artefacts and background noise. Details regarding the platform used for each individual metabolite are provided as Supplementary Data 19 and 20. A total of 1116 metabolites were measured in the 480 faecal samples, including 850 of known chemical identity used in this study. In blood, a total of 902 metabolites were measured in 859 individuals, 687 of which had known chemical identities. Metabolites were scaled by run-day medians, and log-transformed. Faecal metabolites were further scaled to have mean zero and standard deviation one. Metabolites that were indicated as below detection level (zero) were considered as not available (NA).

**Temporal stability of the metabolic profiles**. The blood metabolomic data used in this study belonged to a larger set of 2070 individuals, with longitudinal measurements up to three time points[33], which we used to assess the blood metabolomic stability over time.

In line with the difference observed between the metagenomic and blood metabolomic data used in this study, where about 90% of our samples were collected no more than 2 years apart (Supplementary Fig. 4), we extracted all the individuals having two measurements within a 2-year time frame ($n = 149$), ensuring that their metabolomic profiles were assessed in the same batch in order to limit potential variability due to batch effects. We then extracted, for each tested metabolite's profile, outliers (values further away than three standard deviations from the dataset mean), scaled the data to have mean zero and standard deviation one, and assessed the intra-individual correlations using the Pearson's $\rho$. To confirm that the observed correlations were not due to chance, we then built 10,000 datasets including 149 randomly paired metabolomic profiles from unrelated subjects extracted from the whole metabolomics dataset, ensuring that each pair was measured in the same batch. We then used the Wilcoxon's test to assess the probability of observing a larger average intra-individual correlation in the 149 individuals with measurements taken 2 years apart compared to that observed the random sets. To further verify the stability of the intra-individual correlation over larger time frames, we evaluated the intra-individual correlation for measurements taken up to 10 years apart (and within the same batch).

**Shared microbiome between unrelated individuals**. For all individuals in our dataset, we codified the absence/presence of a microbial species/metabolic pathways with 0 and 1, respectively. Then, after having identified all possible pairs of unrelated individuals ($n = 1,006,288$), we assessed for each pair the percentage of shared species/pathways as the ratio between the number of species/pathways which were present in both members and the number of species/metabolic pathways which were present in at least one of them. The distribution of percentages obtained for species and pathways across all pairs were then compared using a paired Wilcoxon's test.

**Metagenome-wide association study**. Associations of faecal and blood metabolites with species and microbial metabolic pathways-transformed relative abundances were carried out using PopPAnTe (v, 1.0.2)[12], which uses a variance component framework and the matrix of the expected kinship between each pair of individuals, generated using the pedigree information, to model the resemblance between family members. Sex and age at the sample collection were included as covariates. Only pairs of metabolites–species/pathways with at least 50 observations were tested for association. In these analyses, we used all the available samples with faecal metabolites ($n = 479$) and with blood metabolites ($n = 859$). The significance of the associations was evaluated by comparing the likelihood of a full model, including the species/metabolic pathways in the fixed effect, and the likelihood of a null model where these effects were constrained to zero. Associations passing a FDR threshold of 5% were considered significant. FDR was evaluated using Storey's method[59].

**Age effect**. To assess the effect of age in our analyses, associations of faecal and blood metabolites with species and microbial metabolic pathways-transformed relative abundances were carried out using PopPAnTe, with only pairs of metabolites–species/pathways with at least 50 observations tested for association. We compared, in the same dataset, the results of two models: one including only sex as covariate, and the other including both sex and age at sample collection. In both models, associations passing an FDR threshold of 5% and having concordant direction of effects in the two experimental settings were considered as unaffected by age.

**Effect of medication**. Self-reported use of antibiotics, PPI drugs, and metformin was available for 411 individuals with metagenomic data. Only 13, 26, and 33 out of these 411 individuals were using metformin, PPI, and antibiotics, respectively, with two individuals taking both metformin and PPI, and two other individuals taking both PPI and antibiotics, while 343 individuals (84%) were not taking any of these medications. We compared the results obtained using an association model that included only sex and age at the sample collection as covariates with those obtained, in the same set of individuals, using an association model which had also information on the use of the three reported drugs (each drug included as fixed effect in the PopPAnTe linear mixed model and coded as: 1 = taking the drug or 0 = not taking the drug). Associations passing an FDR threshold of 5% in both experimental settings and showing concordant direction of effects were considered unaffected by these drugs.

**Enrichment analysis**. Enrichment analysis was performed using the super-pathways annotation provided by Metabolon, Inc. Metabolites were grouped in the following eight super-pathways: amino acid, carbohydrate, cofactors and vitamins, energy, lipid, nucleotide, peptide, and xenobiotics. As done previously[11], enrichment $P$ values were evaluated using the parametric analysis of gene set enrichment (PAGE) algorithm[60] using 10,000 random permutations as implemented in the *piano* R package[61] (v 1.20). The PAGE algorithm, being based on a two-tailed $Z$ score, can evaluate whether each super-pathway is significantly enriched for an increase or a decrease of the amount of metabolites which it includes.

**Linking metabolites to MetaCyc metabolic pathways**. We downloaded from the MetaCyc[62] Web interface (version 22.6) the list of all compounds (univocally identified using the MetaCyc compound identifier, and, when available, the InChi Key). Then, using the MetaCyc SmartTables function (option: pathways of compound; https://metacyc.org/PToolsWebsiteHowto.shtml#TAG:__tex2page_sec_6), we generated a table assigning each compound to the pathways they belonged to. Finally, for all the metabolites associated to at least one pathway in faeces and/or blood, we generated a second table listing their InChi Key, when known. We were able to annotate 627/679 and 198/222 faecal and blood metabolites, respectively. An inner joint of the two tables, using the InChi Key as key, highlighted that 155 and 42 of the faecal and blood metabolites annotated in the previous step (and involved in 4891 and 419 unique associations, respectively) were assigned to at least one of the MetaCyc metabolic pathways. This table was used to evaluate the proportion of metabolites associated to pathways that also included the metabolites as substrate or product.

**Gut–host metabolic dialogue**. We selected all pairs of metabolites that were observed in at least 100 individuals and were associated with the same species/metabolic pathway in both environments (co-associated metabolites) and used two

approaches to detect and validate the presence of an interplay between the gut and the systemic host metabolism.

First, we hypothesised that, if a species (or pathway) is involved in the dialogue between faecal and blood metabolites, these metabolites would be expected to be more strongly correlated in the presence of the species (or pathway) than in its absence. We used the missingness observed in our WMGS data to test this hypothesis. Indeed, while we are not able to measure it, we can confidently assume that a variable proportion of missing data in our dataset are likely to include truly missing species (or pathways). We tested this hypothesis through simulations. We selected all pairs of co-associated metabolites interacting with species (or pathways) with at least 30 missing observations, and built 1,000 random datasets which included 1,000 pairs of metabolites matched by correlation and sample size to the original set of co-associated metabolites. These new pairs were then combined with species (or pathways) having the same missingness pattern of the actual associated species (or pathways). Then, we used these simulated datasets to assess the probability of observing increased correlation between metabolites when the species (or pathway) was present in the co-associated metabolites compared to the matched pairs.

Second, we evaluated, for each pair of co-associated metabolites, its $P$-gain statistic (Supplementary Fig. 7), which allows determining whether the ratio between the two metabolites is more informative than the single metabolites alone, therefore suggesting the presence of a relationship between them[15]. To this aim, we first evaluated the log ratios between each pair of co-associated metabolites. Then, we associated the single metabolites and the obtained ratios with the specific species/pathway by fitting a linear mixed effect model in R (package lme4, v. 1.1.18), including age and sex as fixed effects, and family structure as a random effect. All association tests were carried out between pairs of co-associated metabolites and metagenomic data with at least 100 complete observations (i.e., having metagenomic data and metabolic profile for both co-associated metabolites available). Finally, we evaluated the $P$-gain statistic as the ratio between the minimum $P$ value obtained using the single metabolites alone and the $P$ value obtained using their ratio[15]. It has previously suggested that a critical $P$-gain threshold taking into account multiple test correction, under the assumption of a type I error rate of 0.05, would be 10 times the number of tests[15]. However, it has also been observed that the magnitude of the $P$-gain statistic can be reduced by the increasing correlation between the metabolites and their ratio, and increased by an increasing sample size[15], two parameters which varied greatly in our dataset. We, therefore, estimated a null distribution empirically using a conservative assumption of no interplay between metabolites associated with different species or pathways. Therefore, we build a null distribution of $P$-gain statistics using 100,000 pairs of randomly selected metabolites which were associated at a 5% FDR with two different species (or pathways) but were matched 1-to-1 by correlation and sample size to the co-associated metabolite pairs. We used the top 5% $P$-gain value as the critical $P$-gain threshold.

**Adiposity phenotypes data and association study.** Subjects were asked to remove their shoes, and height (in cm) was measured using a stadiometer. Weight (in kg) was measured on digital scales. Total and visceral fat mass percentage was determined in 1141 individuals with metagenomic and/or metabolomic data available by dual-energy X-ray absorptiometry (DXA; Hologic QDR; Hologic, Inc., Waltham, MA, USA) whole-body scanning by a trained research nurse. The QDR System Software Version 12.6 (Hologic, Inc., Waltham, MA, USA) was used to analyse the scans. Measurements >3 standard deviations from the dataset mean were excluded from the analysis. To ensure the normality of their distribution, the data were rank-based inverse normalised. Associations with *M. Smithii*, blood and faecal threonate, and 61 faecal metabolites whose dialogue with blood threonate via *M. smithii* was confirmed by the $P$-gain statistic, were carried out by fitting a linear-mixed effect model in R (package lme4, v. 1.1.18), including age and sex as fixed effects, and family structure as a random effect.

**Reporting summary.** Further information on research design is available in the Nature Research Reporting Summary linked to this article.

## Data availability
Data generated during the study are available as Supplementary Data, while a web interface for querying the associations between microbiome and metabolites is available at: http://www.metabgut.org. Results are shown as interactive tables and can also be visualised graphically. Data on TwinsUK twin participants are available to bona fide researchers under managed access due to governance and ethical constraints. Raw data should be requested via our website (http://twinsuk.ac.uk/resources-for-researchers/access-our-data/) and requests are reviewed by the TwinsUK Resource Executive Committee (TREC) regularly. The raw metagenomic sequences are available from the European Nucleotide Archive website (study accession number: PRJEB32731).

## Code availability
The source code used to assess the dialogue between the gut and systemic host metabolism, is available at: https://github.com/alesssia/microbiome_metabolome_interplay.

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

## Acknowledgements

TwinsUK is funded by the Wellcome Trust, Medical Research Council, European Union, the National Institute for Health Research (NIHR)-funded BioResource, Clinical Research Facility and Biomedical Research Centre based at Guy's and St. Thomas' NHS Foundation Trust in partnership with King's College London. M.F. was partially funded by the CDRF. We gratefully acknowledge support provided by the JPI HDHL funded DINAMIC consortium (administered by the MRC UK, MR/N030125/1).

## Author contributions

A.V., C.I.L.R, J.T.B., T.D.S., and M.F. designed the study. W.L., J.C.V., K.E.N. performed the whole shotgun metagenomic sequencing. R.P.M. performed the metabolome profiling. A.V. and T.C.M. carried on the QC of the metagenomics data. E.d.R. offered support during the analysis of metagenomic data. A.V., C.I.L.R., N.R., and M.F. conducted the computational experimentation. F.R. developed the Web portal. A.V., C.I.L.R., T.D.S., and M.F. drafted the manuscript. All authors read and approved the final draft of the manuscript.

## Competing interests

The following authors are or were employees of Human Longevity, Inc.: W.L., J.C.V., K.E.N. R.P.M is a current employee of Metabolome, Inc. E.d.R. is a current employee of Sanofi. T.D.S. is a consultant for Zoe Global Ltd. The other authors declare that they have no competing interests.
