## [Peer Review File · Nature Communications]

Reviewers' comments:

Reviewer #1 (Remarks to the Author):

In the present manuscript, Dr Alessia Visconti et al. confirm greater similarity for functional potentials of metabolic pathways than for taxonomic compositions across shotgun-sequenced metagenomics of stool-derived microbial DNA from a mixed group (predominantly middle-aged and elderly women) of 322 monozygotic twins, 402 dizygotic twins and 280 singletons from UK.

In subsets of the individuals (n = 479) where untargeted metabolomics profiling of faecal water was available, the investigators confirm their previously reported study that the faecal metabolome to some extent reflect the gut microbial activity. In individuals where both metagenomics as well as faecal and blood metabolomics data were available the authors describe various interesting correlations between the three omics profiles. Also, by applying P-gain statistics they identify six key bacterial species that are likely to play a role in mediation of cross-talks between faecal and blood metabolites. Additionally, the authors report correlations between methanogens, blood threonate and three measures of host adiposity.

Comments

The microbiome part of the project are undertaken with standard methods. However, there are several limitations of applied microbiome data including no information on adjustment for variation in stool consistency, sample variation of bacterial cell counts or adjustment for drug intake.

From studies of large-scale populations like the US Human Microbiome Project it has already been reported, that the gut microbiota is relatively more shared among individuals at the functional potential level than at the taxonomic level [doi:10.1038/nature11550]. Thus, the metabolic pathways similarity was firstly reported in a study of 18 females who shared more than 93% of the enzyme-level functional groups. The observation was validated and extended in the much larger populations in MetaHIT [Nature 464, 59-65 (2010)]. The first part of the present results are therefore a further validation of important observations.

Also as emphasized in a previous report by the the TwinsUK investigators the faecal metabolome does reflect faecal microbiome activity to a considerable extent [doi: 10.1038/s41588-018-0135-7]. As such, there is no doubt that carefully conducted faecal metabolic profiles act as a proxy for a functional readout of the gut microbiome. Hence, in part two of the results, the authors extend their previously reported findings.

Age is a known regulator of the faecal microbiome and metabolome [doi: 10.1038/s41588-018-0135-7]. In the actual study, the authors do not give detailed age distribution of all the enrolled participants. The paper will benefit from analyses of the effects of age on the microbiome-fecal metabolome-blood metabolome interplays.

The multiple correlations between the microbiome and two-compartments metabolomes are of interest. Yet, what matters for human biology is to promote novel insights between the intestinal microbiome, the various biological compartments metabolomes and the human host physiology as for instance has been done by other investigators (PMID: 30382244 or PMID: 30401435). Apart from correlations between microbial features and a vitamin C metabolite and host adiposity measures, the authors do not expand their analyses to host nutrient metabolism or hormone actions. The paper will gain in originality if such analyses are done and added.

The commercial approach applied for metabolomics is a considerable concern. This reviewer strongly recommend the authors to add detailed information about quality assured methods used for the metabolomics profiling to the supplementary information. Otherwise, it will not be possible for independent investigators to reproduce the present findings when applying comparable non-commercial methodologies. For the current version of the manuscript, it is not possible to give a balanced judgement of the quality and accuracy of the metabolomics methodologies. When finally reported the metabolomics data needs to be deposited in the MetaboLights database or similar public databases.

Obviously, a particular challenge in metabolomics is the interpretation of the large amount of data. The present study lacks adequate information on statistical procedures to permit replication. It is not clear in what order and to which metabolites each method was applied.

Another major issue in untargeted metabolomics is the accuracy of quantitation. In untargeted profiling, the analytes eluting from the LC system must be ionized to allow their transition from the liquid phase into the gas phase before they can enter the orifice of the mass spectrometry. Among other factors, such as the structural composition of the analyte, the efficiency of the ionization process is dependent on co-eluting metabolites. High-abundant metabolites may suppress the ionization of low-abundant metabolites by picking up most of the available charges at the cone of the spray tip. Thus, the signal intensity is not solely dependent on the concentration of the metabolite. The ionization efficiency, on the other hand, is in part determined by the "matrix". In Supplement, the authors are encouraged to deliver robust evidence showing how to avoid the influence from co-eluting metabolites or contaminants on the quantitation accuracy for the presented analytes.

If the authors performed accurate quantification by generating calibration curves of areas-under-the-curve for peaks against serial amount of reference standards for metabolites, the accurate

concentrations should be given as numbers followed by units. If not, this reviewer recommends to use abundance (without units) instead of concentration.

Reviewer #2 (Remarks to the Author):

Visconti et al. present a concise manuscript that reports on the association of faecal and blood metabolites with gut microbial taxa and their metabolic repertoire. The manuscript is well written. The authors chose a small but efficient set of tools to analyse the microbiome and evaluate associations. The manuscript is however weak on the interpretation of the results with respect to microbial and human physiology. There are some critical omissions of methods, ethics and data statements. Overall, the manuscript does provide compelling evidence for the appropriateness of functional analyses of the human microbiome in health in disease, but would strongly profit from addressing the points detailed below.

Major points:

- There is a strong focus of this manuscript on the greater similarity of pathways across individuals compared to taxonomic profiles. This is probably the least novel aspect in the findings of this manuscript and I would advice to abbreviate its presentation in favor of a more balanced discussion of its representation in published literature and the clarification of the points below that are more novel and less intuitively grasped.

- The authors interpret their results to indicate that “coordinated action of multiple taxa is required to affect the metabolome”. This would suggest some kind of cross-feeding, which certainly exists in the human microbiome. However, the observed patterns could also be due to functional redundancy and are probably more easily explained as such. In particular, the approach chosen by the authors to test for the presence of pathways in the microbiome only takes into account (near-to) complete pathways, indicating that intra-microbiome interactions that rely on cross-feeding may be overlooked in the analysis.

I would urge the authors to include this alternative interpretation, potentially replacing the less likely currently presented view.

- There is no ethics statement.

- There is no data availability statement for the raw data.

- I am missing a discussion on the functional relevance of the observed associations. For example, sebacate, is this rather unusual metabolite assumed to be produced by gut microbes? Are any of the metabolites related to pharmaceuticals that could also affect the microbiome? In the discussion, it is very briefly mentioned that the associations are often between pathways and unrelated metabolites. I would like to know more about this in the results section. What proportion of associations with metabolites that appear in the pathway (or superpathway) and how many are unexplained associations? on a related note, it should be explicitly mentioned already in the results section that the associations may be positive or negative and potentially which proportion has which direction. On the other hand, I am missing some obvious candidates for microbial interaction, namely the bile acids. Were these not as commonly associated? or not measured? A discussion of this would be helpful. Another point with relevance to the interpretation of the data is in the discussion "90% of the microbial species interact with their surrounding metabolic environment" - I fail to see how the other 10% survive.

Further points:

In the abstract, it would be helpful to mention which similarity measure is being applied.

Similarly, it would be helpful to mention the method employed for association testing in the results section and the legend of figure 2. It may be obvious to researchers from the human genetics field, but after reading the presentation of results from unrelated subjects in the first part of the results section, I didn't realize that the association analysis actually uses kinship information. On a related note, the methods section should state whether pedigree information or genomic data was used in the association models.

The methods section is lacking details on sampling, and especially storage conditions and duration. It is also unclear whether faecal samples for metagenomics and metabolomics are the same samples

or were taken at different points in time. Similarly, it is unclear whether the blood samples were taken at the same time.

Numbers of reads: an average of 39 M QC'ed and filtered reads is mentioned in the abstract and the results section. However, in the methods section, it is mentioned that roughly 27M reads were sequenced per samples were sequenced. These numbers don't add up.

Figure 2: the N stated at the top of the figure is misleading. Indeed, none of the displayed analyses were performed with 1004 individuals, as the original cohort was larger and the cohorts used for the two analyses are both considerably smaller. The 1004 should be replaced with the sample numbers for both arms.

On a related note, for the comparisons of associations in blood and faecal samples, were all results used or only those relating to the cohort that had faecal and blood metabolites measured?

The differences of correlations in the absence or presence of species could be due to common reasons for correlation and species presence. Eg. if metabolite comes from a food source that also allows for the microbial species' growth. This test for differences in correlation therefore is no proof of interaction.

Figure 3: the arrows for the bile acids are a bit confusing. Bile acids aren't produced in the blood, as is suggested, and they are transformed by the microbiome, so the arrows shouldn't end at the microbiome.

The discussion lists some limitations of the presented study. I would argue that another limitation is that the microbiome analyses are not quantitative, i.e. there is no estimate of microbial density and therefore it is impossible to say if a relative abundance of one pathway in one sample is more or less than the abundance in another sample.

Language:

throughout:

- I would replace "gut lumen metabolic content" (and similar) by "faecal metabolic content", because that is where the metabolites were measured. Also, gut lumen could refer to different gut sections and is therefore less precise.

- there are several instances where the authors refer to pathways or species “affecting” or “controlling” metabolite abundances. Since the analyses are associative, these terms are over-interpretative and should (at least in the results section) be replaced by more appropriate choices.

Intro, paragraph 1:

- “variations in its composition induces” should be “induce”

Intro, paragraph 2:

- “16S amplicon data” should be “16S rRNA gene amplicon data” (or “16S rRNA amplicon data”, if it refers to RNA-based assays)

- “host’s health” should be “hosts’ health” or “host health”

Gut microbiota composition is host-specific whereas its functions are shared across subjects:

par 1:

- should be “Subdoligranulum genus”

- “Microbial metabolic profiling” is a bit misleading. “Microbial metabolic pathway detection” might be more to the point.

par. 2:

- “pathway prevalence within our sample strongly correlated with the number of species in which it could be detected” - does the detection refer to the same sample or the overall dataset?

The microbiome is involved in the crosstalk between the gut lumen and host systemic metabolism:

- In my opinion, the term “crosstalk” is inadequate here, as it indicates some kind of signalling process. It would probably help to define already here what kind of mechanisms could lead to the observed patterns and choose a term that reflects them.

par. 1, last sentence - introduce singular article for “species” or change “was” to “were”

par. 2, 3rd sentence - “two metabolite levels” would be more clear than “two metabolites”, since this can refer to the same metabolite in two matrices.

Discussion - end of par. 3 - are the associations with vitamin B metabolites in blood or faeces or both?

Reviewers' comments:

Reviewer #1 & Reviewer #2

The multiple correlations between the microbiome and two-compartments metabolomes are of interest. Yet, what matters for human biology is to promote novel insights between the intestinal microbiome, the various biological compartments metabolomes and the human host physiology as for instance has been done by other investigators (PMID: 30382244 or PMID: 30401435).

Apart from correlations between microbial features and a vitamin C metabolite and host adiposity measures, the authors do not expand their analyses to host nutrient metabolism or hormone actions. The paper will gain in originality if such analyses are done and added.

Visconti et al. present a concise manuscript that reports on the association of faecal and blood metabolites with gut microbial taxa and their metabolic repertoire. The manuscript is well written. The authors chose a small but efficient set of tools to analyse the microbiome and evaluate associations. The manuscript is however weak on the interpretation of the results with respect to microbial and human physiology. There are some critical omissions of methods, ethics and data statements. Overall, the manuscript does provide compelling evidence for the appropriateness of functional analyses of the human microbiome in health in disease, but would strongly profit from addressing the points detailed below.

Answer: We acknowledge that our manuscript places limited focus on physiological impact, apart from a number of examples inspired by the top results from our analyses. This is due to the fact that the goal of our study was to investigate, in human and at a very high level of detail, the interplay between the metagenome and the microbial and host metabolisms in order to a) understand which faecal and blood metabolites associate with the abundances of microbial species and metabolic pathways, and, particularly, b) to identify trios of microbiota/faecal metabolite/blood metabolite that are likely to be interconnected, thus highlighting potential interactions between the systemic and faecal environments.

In this study we present several thousand (39,074) novel associations and potential interconnections between the microbiome and both the gut and the systemic metabolic environments, many of which may be involved in disease risk. Therefore, while interesting, we believe that focusing the manuscript on a particular disease or trait would obfuscate the main scope of this research, that is, to make available to the scientific community a new resource providing a wide picture of the dialogue between the microbiome and the faecal and blood metabolome, which has never been explored at this level of detail in a large human dataset. Our findings are made fully available through extensive Supplementary Materials and through a Web portal (<http://173.212.245.153:3838/hli>) where they can be queried and visualised both graphically and as interactive tables. These results provide scientists with a unique resource that will, for example, help in pinpointing potential biomarkers and targets capable of modulating

the abundances of metabolites and of species and functions relevant for human health for further investigations, or inform microbiology research on potential new metabolic functions of the gut microbiome. While we strongly agree with the Reviewer that the results of our study provide a large resource permitting future integration with physiological phenotypic data, we feel that such an investigation is beyond the scope of the current study.

We now emphasised this in the manuscript by adding to the Discussions:

“Moreover, with this study, we make available to the scientific community a unique resource providing a detailed investigation of the dialogue between the microbiome and the faecal and blood metabolome, which will help in pinpointing potential biomarkers and targets capable of modulating the abundances of metabolites and of species and functions relevant for human health for further investigations, and inform microbiology research on potential new metabolic functions of the gut microbiome. Results are made fully available through extensive Supplementary Materials and through a Web portal (<http://173.212.245.153:3838/hli>) where they can be queried and visualised both graphically and as interactive tables.”

and to the abstract:

“This study provides the scientific community with an unique resource describing the dialogue between the microbiome and the systemic and faecal metabolic environments.”

From studies of large-scale populations like the US Human Microbiome Project it has already been reported, that the gut microbiota is relatively more shared among individuals at the functional potential level than at the taxonomic level [doi:10.1038/nature11550]. Thus, the metabolic pathways similarity was firstly reported in a study of 18 females who shared more than 93% of the enzyme-level functional groups. The observation was validated and extended in the much larger populations in MetaHIT [Nature 464, 59-65 (2010)]. The first part of the present results are therefore a further validation of important observations.

There is a strong focus of this manuscript on the greater similarity of pathways across individuals compared to taxonomic profiles. This is probably the least novel aspect in the findings of this manuscript and I would advice to abbreviate its presentation in favor of a more balanced discussion of its representation in published literature and the clarification of the points below that are more novel and less intuitively grasped.

Answer:

We agree, the first part of our results is indeed a validation and refinement of previous findings. Nonetheless, we provide here novel estimates, on a large dataset, of the metabolic pathways vs species similarity across and between unrelated subjects. We have now amended the first paragraph of the Discussions as follow:

“A previous report on a small sample of female subjects (N = 18) showed that, despite a high β -diversity at the phyla level, between 26-53% of the ‘enzyme’-level functional groups were shared among samples¹⁹. Higher similarity of microbial metabolic pathways vs organismal abundances was also observed by the larger (N=242) Human Microbiome Project²⁰. This may

be explained by the high functional redundancy of pathways across different microbial species²¹. Our larger study validates these findings, and estimates that 12% of the microbiome metabolic potential (as described by the MetaCyc microbial metabolic pathways) is present in all individuals. More in general, a random pair of unrelated subjects shares on average 82% of their microbial metabolic pathways, while this is the case for only 43% of the species.”

We also edited the abstract as:

“We observed that a random pair of unrelated subjects shares, on average, a far greater number of functional metabolic pathways (82%) than of species (43%).”

Reviewer #1

In the present manuscript, Dr Alessia Visconti et al. confirm greater similarity for functional potentials of metabolic pathways than for taxonomic compositions across shotgun-sequenced metagenomics of stool-derived microbial DNA from a mixed group (predominantly middle-aged and elderly women) of 322 monozygotic twins, 402 dizygotic twins and 280 singletons from UK. In subsets of the individuals (n = 479) where untargeted metabolomics profiling of faecal water was available, the investigators confirm their previously reported study that the faecal metabolome to some extent reflect the gut microbial activity. In individuals where both metagenomics as well as faecal and blood metabolomics data were available the authors describe various interesting correlations between the three omics profiles. Also, by applying P-gain statistics they identify six key bacterial species that are likely to play a role in mediation of cross-talks between faecal and blood metabolites. Additionally, the authors report correlations between methanogens, blood threonate and three measures of host adiposity.

The microbiome part of the project are undertaken with standard methods. However, there are several limitations of applied microbiome data including no information on adjustment for variation in stool consistency, sample variation of bacterial cell counts or adjustment for drug intake.

Answer: We agree with the Reviewer's concerns. Unfortunately, neither stool consistency nor bacterial cell counts were collected for this dataset. We now included this limitation in the Discussions Section:

"Forth, stool consistency and microbial cell count, which can have an influence on the gut microbiota composition^{49,50}, were not recorded in this study."

Out of 1,004 individuals with metagenomics data, 411 of them self-reported whether they were taking antibiotics, proton proton inhibitors (PPI), and/or metformin, which have all been shown to have a strong effect on the metagenome composition [DOI: 10.1016/j.cell.2012.10.052, DOI: 10.1038/nature15766, DOI: 10.1136/gutjnl-2015-310861]. Only 13, 26 and 33 out of these 411 individuals were using metformin, PPI, and antibiotics, respectively, with two individuals taking both metformin and PPI, and two other individuals taking both PPI and antibiotics, while 343 individuals (84%) were not taking any of these medications.

We believe that the resulting sample is too small to understand the influence of drug intake on the observed associations between the gut microbiota and both faecal and blood metabolites. Nonetheless, we run a new analysis where the intake of any of these these drugs was included as confounder, and compared the obtained results to those obtained, in the same subset of samples, when the drug intake was not taken into account (Supplementary Table 3, Supplementary Data D10-D13). We obtained very similar results, observing correlations of the betas from association larger than 0.95 ($P < 2.2 \times 10^{-16}$) for all experimental settings. All associations concordantly showed the same direction, while the overlap of those that passed threshold for significance after multiple testing correction (FDR=5%) was between 87-98%.

We have now added these results in the manuscript, but leave it to the judgement of both Editor and Reviewer whether they are interesting enough to be included in the manuscript, or whether they should be better removed.

We add in the Results Section:

*“Similarly, correction for drug intake (antibiotics, metformin, and proton-proton inhibitor -- PPI), which was assessed in a small subset of our study sample (N=411, **Methods**), appeared to minimally affect the number of significant associations between the metagenome and the faecal and blood metabolome (**Supplementary Table 3, Supplementary Data D10-D13**).”*

in the Discussions:

“We also observed, in a small subset of our study sample, that antibiotics, metformin, or PPI intake had a minimal effect on the associations between the microbiome and the metabolome, although this is likely due to the limited number of individuals taking any of these three drugs.”

and in the Methods:

“Effect of medication

Self-reported use of antibiotics, proton-proton inhibitor (PPI) drugs, and metformin was available for 411 individuals with metagenomic data. Only 13, 26 and 33 out of these 411 individuals were using metformin, PPI, and antibiotics, respectively, with two individuals taking both metformin and PPI, and two other individuals taking both PPI and antibiotics, while 343 individuals (84%) were not taking any of these medications. We compared the results obtained using an association model that included only sex and age at the sample collection as covariates with those obtained, in the same set of individuals, using an association model which had also information on the use of the three reported drugs (each drug included as fixed effect in the PopPAnTe linear mixed model and coded as: 1 = taking the drug or 0 = non taking the drug). Associations passing an FDR threshold of 5% in both experimental settings and showing concordant direction of effects were considered unaffected by these drugs.”

Supplementary Table 3. Number of associations observed, in a subset of 411 individuals, when not correcting for use of antibiotics, metformin and PPI at sample collection, and number of associations that remained significant (5% FDR), showing also concordant direction of effects, when the information on these three drugs was taken into account.

		Without correction for drug intake	Overlap N (%)
Faeces	Species	1000	917 (91.7%)
	Pathways	6292	5847 (92.9%)
Blood	Species	43	42 (97.7%)
	Pathways	569	498 (87.5%)

Also as emphasized in a previous report by the the TwinsUK investigators the faecal metabolome does reflect faecal microbiome activity to a considerable extent [doi: 10.1038/s41588-018-0135-7]. As such, there is no doubt that carefully conducted faecal metabolic profiles act as a proxy for a functional readout of the gut microbiome. Hence, in part two of the results, the authors extend their previously reported findings.

Answer: We agree with the Reviewer. Indeed, in our study we extend Zierer's work [DOI: 10.1038/s41588-018-0135-7] by a) identifying association at the species level (whereas Zierer *et al*'s study, based on 16S rRNA gene amplicon data, was limited to the genus level), and b) studying the association with the MetaCyc microbial metabolic pathways inferred from the metagenomics data. We now extend the Discussions as:

“The results at the taxonomic level were comparable to those previously reported in a recent study on the TwinsUK cohort leveraging 16S rRNA gene amplicon data¹¹. In both studies, we observed that over 90% of microbes were associated with a vast proportion of the measured gut metabolites (>80%). The WMGS data used in this study allowed us to extend these observations, by improving the precision of the taxonomic associations at the species level rather than at the genus level. [...] Additionally, the WMGS data allowed the inference of microbial metabolic pathways and their association with the faecal metabolome, which could not be performed on the previous TwinsUK study.”

Age is a known regulator of the faecal microbiome and metabolome [doi: 10.1038/s41588-018-0135-7]. In the actual study, the authors do not give detailed age distribution of all the enrolled participants. The paper will benefit from analyses of the effects of age on the microbiome-faecal metabolome-blood metabolome interplays.

Answer: We agree with the Reviewer regarding the known effect of age on both microbiome and metabolome, which we indeed included as covariate in all of our analyses. Our sample includes predominantly active middle-aged woman, with a mean age of 65.0 ± 7.8 (range: 47.8 - 87.9), as now reported both in Supplementary Table 1 and in Supplementary Figure 5. To assess the effect of age as suggested by the Reviewer, we have now run for comparison the association studies without including age at sample collection as a covariate. We observed that, at 5% FDR, the number of significant associations showing the same directions of effects between both faecal and blood metabolites and the microbiome (both as metabolic pathways and species) was minimally affected by age (Supplementary Table 2), and the correlations between the betas from the linear models, including and not including age, were larger than 0.97 ($P < 2.2 \times 10^{-16}$) for all experimental settings. This is in line with previous observations that show that, while it is well known that the human gut microbiome changes massively during infancy (mostly in the first three years of life, e.g., [DOI: 10.1038/nature11053]), it remains relatively stable across adulthood [DOI: 10.1073/pnas.1423854112, DOI: 10.1126/science.1237439] in absence of external perturbations. We have amended the manuscript accordingly.

Results: *“We observed that age at the sample collection had a negligible effect on the number of significant associations identified between faecal and blood metabolites and both microbial*

metabolic pathways and species (**Methods, Supplementary Table 2, Supplementary Data D6-D9, Supplementary Figure 5**).

Discussions: “In our dataset, composed predominantly of active middle-aged women, we observed that associations between the gut microbiome and both the gut and systemic metabolisms were minimally impacted by age. This is in line with previous observations that show that, in absence of external perturbation, the gut microbiome of healthy adults remains relatively stable for years^{33,34}.”

Methods:

“Age effect.

To assess the effect of age in our analyses, associations of faecal and blood metabolites with species and microbial metabolic pathways transformed relative abundances were carried out using PopPAnTe, with only pairs of metabolites-species/pathways with at least 50 observations tested for association. We compared, in the same dataset, the results of two models: one including only sex as covariate, and the other including both sex and age at sample collection. In both models, associations passing an FDR threshold of 5% and having concordant direction of effects in the two experimental settings were considered as unaffected by age.”

Supplementary Figure 5. Age distribution in the study dataset.

Supplementary Table 2. Number of associations observed when correcting for age at the sample collection, and number of associations that remain significant, at 5% FDR, showing also concordant direction of effects, when age was not taken into account.

		Correcting for age N	Overlap N (%)
Faeces	Species	2493	2237 (89.7%)
	Pathway	16133	15646 (97.0%)
Blood	Species	254	234 (92.1%)
	Pathways	2030	1980 (97.5%)

The commercial approach applied for metabolomics is a considerable concern. This reviewer strongly recommend the authors to add detailed information about quality assured methods used for the metabolomics profiling to the supplementary information. Otherwise, it will not be possible for independent investigators to reproduce the present findings when applying comparable non-commercial methodologies. For the current version of the manuscript, it is not possible to give a balanced judgement of the quality and accuracy of the metabolomics methodologies.

Answer: We have not included a detailed description of the two metabolomic datasets used in our study because both the faecal [DOI: 10.1038/s41588-018-0135-7] and the blood metabolome [DOI: 10.1038/ng.3809] were already described in previous publications from our group. We add them below as they appear in their original manuscripts.

For the faecal metabolome [DOI: 10.1038/s41588-018-0135-7]:

“Sample preparation for global metabolomics

Samples were stored at -80°C until processing. Sample preparation was carried out as described previously at Metabolon, Inc. Lyophilized fecal samples were extracted at a constant per-mass basis. Briefly, recovery standards were added before the first step in the extraction process for quality-control purposes. To remove protein, dissociate small molecules bound to protein or trapped in the precipitated protein matrix, and recover chemically diverse metabolites, proteins were precipitated with methanol under vigorous shaking for 2 min (Glen Mills Genogrinder 2000), then centrifuged. The resulting extract was divided into five fractions: (i) acidic positive-ion conditions chromatographically optimized for more hydrophilic compounds; (ii) acidic positive-ion conditions chromatographically optimized for more hydrophobic compounds; (iii) basic negative-ion-optimized conditions with a separate dedicated C18 column; (iv) negative ionization after elution from a HILIC column; (v) reserved for backup.

Three types of controls were analyzed in concert with the experimental samples: a pooled sample generated from a small portion of each experimental sample of interest served as a technical replicate throughout the platform run; extracted water samples served as process

blanks; and a cocktail of standards spiked into every analyzed sample allowed for instrument performance monitoring. Instrument variability was determined by calculation of the median relative s.d. (RSD) for the standards that were added to each sample before injection into the mass spectrometers (median RSDs were determined to be 5%; n=31 standards). Overall process variability was determined by calculating the median RSD for all endogenous metabolites (i.e., noninstrument standards) present in 90% or more of the pooled technical-replicate samples (median RSD = 12%, n = 832 metabolites). Experimental samples and controls were randomized across the platform run.

Mass spectrometry analysis

Extracts were subjected to UPLC–MS/MS35. The chromatography was standardized, and no further changes were made after the method was validated. As part of Metabolon's general practice, all columns were purchased from a single manufacturer's lot at the outset of experiments. All solvents were similarly purchased in bulk from a single manufacturer's lot in sufficient quantity to complete all related experiments. For each sample, vacuum-dried samples were dissolved in injection solvent containing eight or more injection standards at fixed concentrations, depending on the platform. The internal standards were used to ensure both injection and chromatographic consistency. Instruments were tuned and calibrated for mass resolution and mass accuracy daily.

All methods used a Waters Acquity UPLC and a Thermo Scientific Q-Exactive high-resolution/accurate mass spectrometer interfaced with a heated electrospray ionization (HESI-II) source and an Orbitrap mass analyzer operated at 35,000 mass resolution. The sample extract was dried, then reconstituted in solvents compatible with each of the four methods. Each reconstitution solvent contained a series of standards at fixed concentrations to ensure injection and chromatographic consistency. One aliquot was analyzed by using acidic positive-ion conditions, which were chromatographically optimized for relatively hydrophilic compounds. In this method, the extract was gradient eluted from a C18 column (Waters UPLC BEH C18, 2.1 × 100 mm, 1.7 μm) with water and methanol containing 0.05% perfluoropentanoic acid and 0.1% formic acid. Another aliquot was also analyzed by using acidic positive-ion conditions; however, it was chromatographically optimized for relatively hydrophobic compounds. In this method, the extract was gradient eluted from the same aforementioned C18 column with methanol, acetonitrile, water, 0.05% perfluoropentanoic acid, and 0.01% formic acid, and was operated at an overall higher organic content. Another aliquot was analyzed by using basic negative-ion-optimized conditions and a separate dedicated C18 column. The basic extracts were gradient eluted from the column with methanol and water, as well as 6.5 mM ammonium bicarbonate at pH 8. The fourth aliquot was analyzed via negative ionization after elution from a HILIC column (Waters UPLC BEH Amide 2.1 × 150 mm, 1.7 μm) with a gradient consisting of water and acetonitrile with 10 mM ammonium formate, pH 10.8. The MS analysis alternated between MS and data-dependent MS_n scans using dynamic exclusion. The scan range varied slightly between methods but covered 80–1,000 m/z.

Compound identification, quantification, and data curation

Metabolites were identified by automated comparison of the ion features in the experimental samples to a reference library of chemical standard entries that included retention time, molecular weight (m/z), preferred adducts, and in-source fragments as well as associated MS spectra, and were curated by visual inspection for quality control in software developed at

Metabolon. Identification of known chemical entities was based on comparison to metabolomic library entries of purified standards. Commercially available purified standard compounds have been acquired and registered into LIMS for distribution to the various UPLC-MS/MS platforms for determination of their detectable characteristics. Additional mass-spectral entries have been created for structurally unnamed biochemicals, which have been identified on the basis of their recurrent nature (both chromatographic and mass spectral). These compounds have the potential to be identified by future acquisition of a matching purified standard or by classical structural analysis. Peaks were quantified through area-under-the-curve analysis. Raw area counts for each metabolite in each sample were normalized to correct for variation resulting from instrument interday tuning differences by the median value for each run day, and the medians were therefore set to 1.0 for each run. This procedure preserved variation among samples but allowed metabolites of widely different raw peak areas to be compared on a similar graphical scale.

A total of 1,116 different metabolites were measured in the 786 fecal samples, of which 210 metabolites were observed in less than 20% of the samples and thus were excluded from further analysis because of lack of power. 345 metabolites were observed in more than 20% but less than 80% of the samples and were thus analyzed qualitatively as dichotomous traits (observed in a sample versus not observed). The remaining 570 metabolites, which were observed in at least 80% of all samples, were scaled by run-day medians, log-transformed and scaled to uniform mean 0 and s.d. 1 and analyzed quantitatively (Fig. 1). Metabolite ratios were calculated from the run-day median-normalized metabolite levels and subsequently log-transformed and scaled to a mean of 0 and s.d. of 1.

We analyzed effects of sample storage time (i) in the refrigerator before samples were frozen and (ii) in the freezer before further analysis. To this end, we regressed metabolite concentrations against storage times. After correcting for multiple testing, we found significant storage effects on seven metabolites (FDR <0.05; Supplementary Fig. 5). We thus corrected all further analyses for both storage time in the refrigerator and freezer, to avoid spurious results. Despite correcting all models for the storage time, we cannot ultimately eliminate a potential confounding effect due to storage time, and future studies should investigate the influence of storage time on fecal metabolites.”

For the blood metabolome [DOI: 10.1038/ng.3809]:

“The non-targeted metabolomics analysis was performed at Metabolon (Durham, North Carolina, USA) on a platform consisting of four independent ultra-high-performance liquid chromatography–tandem mass spectrometry (UPLC–MS/MS) instruments. Samples were prepared using the automated MicroLab STAR system from Hamilton Company. Several recovery standards were added before the first step in the extraction process for quality control purposes. To remove protein, to dissociate small molecules bound to protein or trapped in the precipitated protein matrix, and to recover chemically diverse metabolites, proteins were precipitated with methanol under vigorous shaking for 2 min (Glen Mills GenoGrinder 2000) followed by centrifugation. The resulting extract was divided into five fractions: two for analysis by two separate reverse-phase (RP)/UPLC–MS/MS methods with positive-ion-mode electrospray ionization (ESI), one for analysis by RP/UPLC–MS/MS with negative-ion-mode ESI, one for analysis by HILIC/UPLC–MS/MS with negative-ion-mode ESI, and one reserved for

backup. Samples were placed briefly on a TurboVap (Zymark) to remove the organic solvent. The sample extracts were stored overnight under nitrogen before preparation for analysis. Several types of controls were analyzed in concert with the experimental samples: a pooled matrix sample generated by taking a small volume of each experimental sample (or, alternatively, a pool of well-characterized human plasma) served as a technical replicate throughout the data set; extracted water samples served as process blanks; and a cocktail of quality control standards that were carefully chosen not to interfere with the measurement of endogenous compounds was spiked into every analyzed sample, allowed instrument performance monitoring and aided chromatographic alignment. Instrument variability was determined by calculating the median relative standard deviation (RSD) for the standards that were added to each sample before injection into the mass spectrometers. Overall process variability was determined by calculating the median RSD for all endogenous metabolites (i.e., non-instrument standards) present in 100% of the pooled-matrix samples. Experimental samples were randomized across the platform run with quality control samples spaced evenly among the injections. All methods used a Waters ACQUITY ultra-performance liquid chromatographer and a Thermo Scientific Q-Exactive high-resolution mass spectrometer interfaced with a heated electrospray ionization (HESI-II) source and Orbitrap mass analyzer operated at 35,000 mass resolution. The sample extract was dried and then reconstituted in solvents compatible with each of the four methods. Each reconstitution solvent contained a series of standards at fixed concentrations to ensure injection and chromatographic consistency. One aliquot was analyzed using acidic positive-ion conditions, chromatographically optimized for more hydrophilic compounds. In this method, the extract was gradient eluted from a C18 column (Waters UPLC BEH C18–2.1 × 100 mm, 1.7 μm) using water and methanol, containing 0.05% perfluoropentanoic acid (PFPA) and 0.1% formic acid (FA). Another aliquot was also analyzed using acidic positive-ion conditions; however, it was chromatographically optimized for more hydrophobic compounds. In this method, the extract was gradient eluted from the same afore-mentioned C18 column using methanol, acetonitrile, water, 0.05% PFPA and 0.01% FA and was operated at an overall higher organic content. Another aliquot was analyzed using basic negative-ion-optimized conditions on a separate dedicated C18 column. The basic extracts were gradient eluted from the column using methanol and water, however, with 6.5 mM ammonium bicarbonate at pH 8. The fourth aliquot was analyzed via negative ionization following elution from a HILIC column (Waters UPLC BEH Amide 2.1 × 150 mm, 1.7 μm) using a gradient consisting of water and acetonitrile with 10 mM ammonium formate, pH 10.8. The MS analysis alternated between MS and data-dependent MS_n scans using dynamic exclusion. The scan range varied slightly between methods but covered 70–1,000 m/z. Raw data files are archived and extracted as described below. Raw data were extracted, peak identified and quality control processed using Metabolon's hardware and software. These systems are built on a web service platform using Microsoft's .NET technologies, which run on high-performance application servers and fiber-channel storage arrays in clusters to provide active failover and load balancing. Compounds were identified by comparison to library entries of purified standards or recurrent unknown entities. Metabolon maintains a library based on authenticated standards that contains the retention time/index (RI), mass-to-charge ratio (m/z) and chromatographic data (including MS/MS spectral data) on all molecules present in the library. Furthermore, biochemical identifications are based on three criteria: retention index

within a narrow RI window of the proposed identification, accurate mass match to the library ± 10 ppm, and MS/MS forward and reverse scores between the experimental data and authentic standards. MS/MS scores are based on a comparison of the ions present in the experimental spectrum to the ions present in the library spectrum. While there may be similarities between these molecules based on one of these factors, the use of all three data points can distinguish and differentiate biochemicals. More than 3,300 commercially available purified standard compounds have been acquired and registered into LIMS for analysis on all platforms for determination of their analytical characteristics. Additional mass spectral entries have been created for structurally unnamed biochemicals, which have been identified by virtue of their recurrent nature (both chromatographic and mass spectral). These compounds have the potential to be identified by future acquisition of a matching purified standard or by classical structural analysis. A variety of curation procedures were carried out to ensure that a high-quality data set was made available for statistical analysis and data interpretation. The quality control and curation processes were designed to ensure accurate and consistent identification of true chemical entities and to remove those representing system artifacts, misassignments and background noise. Metabolon data analysts used proprietary visualization and interpretation software to confirm the consistency of peak identification among the various samples. Library matches for each compound were checked for each sample and corrected if necessary.”

To clarify this, we extended the Methods Section of our manuscript, and direct the readers towards these references to provide them with the necessary information for potential replication. More in details, the following was added to the Methods Section:

*“Metabolite were measured from fecal samples and blood by Metabolon, Inc., Morrisville, North Carolina, USA, by using an untargeted UPLC-MS/MS platform as previously described^{11,32}. Briefly, faecal samples were lyophilised then extracted at a constant per-mass basis while blood samples were used directly for extraction at a constant per-volume basis. Proteins and other macromolecules were removed using methanol precipitation. Samples were run using four different methods as described by Zierer et al.¹¹ (faecal metabolome) and Long et al.³² (blood metabolome), against three controls (a pooled sample, extracted water -blank- and a cocktail of standards). Metabolites were identified by comparison to a referenced library of chemical standards⁵⁶, and area-under-the-curve analysis was performed for peak quantification and normalised to day median value. To ensure high quality of the dataset, control and curation processes were subsequently used to ensure true chemical assignment and remove artefacts and background noise. Further details to help reproducing the present findings using comparable non-commercial methodologies are available in Zierer et al.¹¹, for faecal metabolome and Long et al.³², for blood metabolome, and in the **Supplementary Methods**.”*

When finally reported the metabolomics data needs to be deposited in the MetaboLights database or similar public databases.

Answer: Data on TwinsUK twin participants are available to bona fide researchers under managed access due to governance and ethical constraints. Raw data should be requested via our website (<http://twinsuk.ac.uk/resources-for-researchers/access-our-data/>) and requests are reviewed by the TwinsUK Resource Executive Committee (TREC) regularly.

Obviously, a particular challenge in metabolomics is the interpretation of the large amount of data. The present study lacks adequate information on statistical procedures to permit replication. It is not clear in what order and to which metabolites each method was applied.

Answer: We are sorry that our experimental design was not clear, and that some information were lacking. While Figure 2 described the main analysis, we have now added a novel Supplementary Figure (Supplementary Figure 6, also below) describing the analysis pipeline for the P-gain analysis. We also hope that the clarifications added in the Methods Section regarding the metabolomics profiling will improve replicability.

Supplementary Figure 6. Data selection process for the P-gain analysis.

Another major issue in untargeted metabolomics is the accuracy of quantitation. In untargeted profiling, the analytes eluting from the LC system must be ionized to allow their transition from the liquid phase into the gas phase before they can enter the orifice of the mass spectrometry. Among other factors, such as the structural composition of the analyte, the efficiency of the ionization process is dependent on co-eluting metabolites. High-abundant metabolites may suppress the ionization of low-abundant metabolites by picking up most of the available charges at the cone of the spray tip. Thus, the signal intensity is not solely dependent on the concentration of the metabolite. The ionization efficiency, on the other hand, is in part determined by the “matrix”. In Supplement, the authors are encouraged to deliver robust evidence showing how to avoid the influence from co-eluting metabolites or contaminants on the quantitation accuracy for the presented analytes.

Answer: The Reviewer is correct that non-targeted metabolomics approaches utilized for discovery work are susceptible to ionization effects derived from co-eluting metabolites

(including highly abundant metabolites). As outlined in the Methods, all sample extracts were processed and analyzed using four different analytical methods, which allowed for cross-platform confirmation of the data.

We now add in the Supplementary Methods the following:

“Co-eluting metabolites

About 60% of metabolites reported by Metabolon are identified and measured across more than one platform (or on the same platform using different quantions), although data from only one of the platforms per metabolite were presented in this manuscript. It is unlikely that a given metabolite would encounter the same influence from co-eluting metabolites, such as ion suppression, on more than one platform. Furthermore, as part of Metabolon's standard QC process, correlations of cross-platform (or cross-library entry) measurements were examined to identify influencing factors. When a poor correlation was identified, data from the suspect platform was not reported back; rather, metabolite data were reported from another platform when at least two other correlated platforms were present.

In some cases, co-eluting metabolites that have identical masses and similar structures are known or suspected to be present in the data set (e.g., 2-hydroxybutyrate and 2-hydroxyisobutyrate or 3-methylglutarate and 2-methylglutarate). These isobaric metabolites are presented in the data tables as "metabolite 1/metabolite 2". These metabolites are not resolvable by the chromatography used for the reported platform and were not measured as resolvable metabolites on the remaining three platforms. The metabolite levels presented, therefore, may represent contributions from one or both of the reported metabolites. Based on findings from many thousands of studies that have been run across the Metabolon platform, these isobaric metabolite entries have been determined to represent a very minor percentage of metabolites reported.”

If the authors performed accurate quantification by generating calibration curves of areas-under-the-curve for peaks against serial amount of reference standards for metabolites, the accurate concentrations should be given as numbers followed by units. If not, this reviewer recommends to use abundance (without units) instead of concentration.

Answer: The Reviewer is correct, and we now replace "concentration" by "abundance" when referring to metabolites in the manuscript.

Reviewer #2

The authors interpret their results to indicate that “coordinated action of multiple taxa is required to affect the metabolome”. This would suggest some kind of cross-feeding, which certainly exists in the human microbiome. However, the observed patterns could also be due to functional redundancy and are probably more easily explained as such. In particular, the approach chosen by the authors to test for the presence of pathways in the microbiome only takes into account (near-to) complete pathways, indicating that intra-microbiome interactions that rely on cross-feeding may be overlooked in the analysis. I would urge the authors to include this alternative interpretation, potentially replacing the less likely currently presented view.

Answer: We agree that the observed patterns could be due to functional redundancy, and we now emphasise this interpretation in the Discussions. On the other hand, the fact that most of the metabolic pathways are associated with metabolites apparently unrelated to their functions may suggest that functional redundancy is not the only possible explanation. Since characterising intra-microbiome interactions relying on cross-feeding is not possible in our analyses, we downplay this alternative explanation in the Discussions by writing:

“We suggest that this large number of associations with metabolic pathways is likely due to functional redundancy. Nonetheless, the majority of the metabolic pathways, especially in faeces, were associated with metabolites apparently unrelated to their functions, with only 20 and 44% of the faecal metabolite-pathway associations and blood metabolite-pathway associations linking metabolites with the MetaCyc metabolic pathways either producing or consuming them. Therefore, we cannot exclude that part of the observed associations with pathways are driven by the concerted action of microbial sub-communities rather than only by the specific function of the pathways.”

We also adjusted the abstract by replacing “*the coordinated action of multiple taxa is required to affect the metabolome*” by: “*the action of multiple taxa is required to affect the metabolome*”

There is no ethics statement.

There is no data availability statement for the raw data.

Answer: We acknowledge that this important aspect has been overlooked. We now write in the manuscript (Methods Section):

“St. Thomas’ Hospital Research Ethics Committee approved the study, and all twins provided informed written consent.”

and in the Data availability Section:

“Data on TwinsUK twin participants are available to bona fide researchers under managed access due to governance and ethical constraints. Raw data should be requested via our website (<http://twinsuk.ac.uk/resources-for-researchers/access-our-data/>) and requests are reviewed by the TwinsUK Resource Executive Committee (TREC) regularly. The raw metagenomic sequences are available from the European Nucleotide Archive website (study accession number: PRJEB32731).”

I am missing a discussion on the functional relevance of the observed associations. For example, sebacate, is this rather unusual metabolite assumed to be produced by gut microbes?

Answer: Sebacate is indeed an unusual metabolite, and the current literature on its biological role and provenance is very limited. We have now included the following part in the Discussions: *“Sebacate was the faecal metabolite that associated with the greatest number of species and metabolic pathways. Sebacate metabolism has been poorly studied. However, a pharmacokinetic study of sebacate in rats has revealed, post-ingestion, a low systemic bioavailability, suggesting that this may be explained by direct beta-oxidation of sebacate (i.e., sebacate degradation) by the liver, and that only traces of the compound could be detected in faeces³⁸. Another study on rats also revealed the absence of sebacate in faeces after intravenous injection of the radioactive compound³⁹, indicating that it is unlikely that systemic sebacate level affects the gut microbiome through its excretion in the gut. Sebacate can be used as primary carbon source by some gut commensals (*Pseudomonas aeruginosa* and *Pseudomonas multivoran*)⁴⁰. Thus, the observed low post-ingestion level of sebacate in both faeces and blood in rats, and the numerous associations identified by our study between faecal and blood sebacate and the gut microbiome may also be due to its utilisation by gut bacteria as carbon source. Endogen sebacate, naturally found in blood, can be synthesised, in rats, through omega-oxidation in starvation periods, before undergoing beta-oxidation to produce succinate and be used as energy source through neoglucogenesis^{41,42}. It was also reported that gut bacteria may affect liver beta-oxidation through modulation of the immune system in mice⁴³. Therefore, an alternative/complementary hypothesis might be that the high number of associations observed between blood sebacate and the gut microbiome might picture the effect that the gut microbiome exerts on liver functions⁴⁴.”*

Are any of the metabolites related to pharmaceuticals that could also affect the microbiome?

Answer: This is an interesting point, since drugs may affect both the metabolic activity of the gut microbiome and its composition. To assess this, we followed two approaches. First, we run a novel analysis where the use of metformin, proton-proton inhibitor (PPI), and antibiotics (which was available for 411 individuals belonging to our study sample) was taken into account, and compared the obtained results to those obtained, in the same subset of samples, when the use of these drugs was not taken into account. This approach and its results are detailed in the answer to the first point raised by Reviewer #1.

Additionally, we checked the associations between the microbiome and drugs and drug-related metabolites characterised through the Metabolon platform to understand whether drugs and drug-related metabolites detected in the faeces were associated with changes in the gut microbiome composition and activity. Of the 82 drugs and drug-related metabolites detected by the Metabolon platform, 11 were available for a sufficient number of samples (N≥50). Using these eleven metabolites in faeces, we observed, at FDR 5%, six associations between six

species and three metabolites, and 101 associations between 82 microbial metabolic pathways and six metabolites.

We have added to the Results Section:

*“Eleven of the 82 drug or drug-derived metabolites detected by the Metabolon platform in faeces were present in at least 50 samples with matching metagenomics data. At the species level, we observed six associations with three of these metabolites passing an FDR threshold of 5% (**Supplementary Data D2**). One association was between 3-hydroxyquinine (a degradation product of quinine, used against malaria, but also contained as a flavouring in beverages, including tonic water) and unclassified *Anaerotruncus* spp. ($\beta=0.68$, $SE=0.18$, $P=4.02\times 10^{-5}$). Two negative associations were identified between salicylic acid (a precursor of aspirin) and *Methanobrevibacter smithii* ($\beta=-0.53$, $SE=0.17$, $P = 2.21\times 10^{-4}$) and unclassified *Anaerotruncus* spp. ($\beta=-0.62$, $SE=0.17$, $P=2.13\times 10^{-4}$). Finally, N-carbamylglutamate (a drug that can be used for the treatment of hyperammonemia) was associated with *F. prausnitzii* ($\beta=0.68$, $SE = 0.18$, $P = 2.21\times 10^{-4}$), *Odoribacter splanchnicus* ($\beta=0.88$, $SE=0.25$, $P=3.93\times 10^{-4}$), and *Blautia hydrogenotrophica* ($\beta=-0.57$, $SE=0.15$, $P = 1.17\times 10^{-4}$). At FDR 5%, a total of 101 associations were observed between microbial metabolic pathways and faecal metabolites annotated as drugs or and drug-derived metabolites (**Supplementary Data D3**). Namely: 3-(N-acetyl-L-cystein-S-yl) acetaminophen (26 associations, metabolite derived from paracetamol), 3-hydroxyquinine (1 association), 4-acetamidophenol (24 associations, metabolite derived from paracetamol), carboxyibuprofen (2 associations, metabolite derived from ibuprofen), N-carbamylglutamate (8 associations) and salicylic acid (40 associations).”*

We add the following sentences to the Discussions:

“Drugs can be metabolised by the gut microbiota, and they may affect both the metabolic activity of the gut microbiome and its composition^{27,28}. In our analyses, we identified associations between six species and 101 microbial metabolic pathways and six out of eleven drugs and drug-related metabolites detected in faeces through the Metabolon platform in a sufficient number of subjects.”

In the discussion, it is very briefly mentioned that the associations are often between pathways and unrelated metabolites. I would like to know more about this in the results section. What proportion of associations with metabolites that appear in the pathway (or superpathway) and how many are unexplained associations?

Answer: We thank the Reviewer for making this point. This piece of information was indeed overlooked in the Results Section. We now provide more detailed information on the number of metabolites included in the pathways they are associated to. The MetaCyc databases was queried via the SmartTables function (“pathways of compound” option) in order to assign compounds to pathways. We then annotated the metabolites using the InChi Keys, which were available for 627/679 and 198/222 faecal and blood metabolites with at least one significant association in faeces and blood, respectively. In faeces, 155 out of the 627 annotated faecal metabolites were assigned to at least one of the MetaCyc metabolic pathways. These 155 metabolites were involved in 4,891 unique metabolite-pathway associations, of which 20%

(N=999) encompassed a metabolite included in the associated pathway (20%). In blood, 42 out of the 198 annotated metabolites were assigned to at least one of the MetaCyc metabolic pathways. These 42 metabolites were involved in 419 unique metabolite-pathways associations, 186 of which encompassing a metabolite included in the associated pathway (44%).

We now write in the Results Section:

“Notably, in both faeces and blood, the majority of the metabolic pathways were associated with metabolites apparently unrelated to their functions. Indeed, only 999 out of 4,891 unique faecal metabolite-pathway associations (20%) and 186 out of 419 unique blood metabolite-pathway associations (44%), respectively, linked 155 faecal and 42 blood metabolites to pathways either producing or consuming them (Methods).”

We also include the following paragraph to the Discussions:

“Nonetheless, the majority of the metabolic pathways, especially in faeces, were associated with metabolites apparently unrelated to their functions, with only 20 and 44% of the faecal metabolite-pathways associations and blood metabolite-pathways associations linking metabolites with the MetaCyc metabolic pathways either producing or consuming them”.

Finally, we extended the Methods as:

“Linking metabolites to MetaCyc metabolic pathways

We downloaded from the MetaCyc⁶¹ Web interface (version 22.6) the list of all compounds (univocally identified using the MetaCyc compound identifier, and, when available, the InChi Key). Then, using the MetaCyc SmartTables function (“pathways of compound” option; https://metacyc.org/PToolsWebsiteHowto.shtml#TAG:tex2page_sec_6), we generated a table assigning them to the pathways they belonged to. Finally, for all the metabolites associated to at least one pathway in faeces and/or blood, we generated a second table listing their InChi Key, when known. We were able to annotate 627/679 and 198/222 faecal and blood metabolites, respectively. An inner joint of the two tables, using the InChi Key as key, highlighted that 155 and 42 of the faecal and blood metabolites annotated in the previous step (and involved in 4,891 and 419 unique associations, respectively) were assigned to at least one of the MetaCyc metabolic pathways. This table was used to evaluate the proportion of metabolites associated to pathways that also included the metabolites as substrate or product.”

on a related note, it should be explicitly mentioned already in the results section that the associations may be positive or negative and potentially which proportion has which direction.

Answer: We agree with the Reviewer’s comment and added the percentage of positive associations in the Results Section. For faecal:

“We observed 48% and 51% positive associations with microbial metabolic pathways and species, respectively.”

For blood:

“At a 5% FDR, we identified 2,030 associations with microbial metabolic pathways and 254 associations with microbial species, of which 44 and 43% were positive, respectively.”

On the other hand, I am missing some obvious candidates for microbial interaction, namely the bile acids. Were these not as commonly associated? or not measured? A discussion of this would be helpful.

Answer: We agree with the Reviewer that the interaction between bile acids and the gut microbiome has been extensively described in the literature and worth discussing. Bile acids were measured in both blood and faeces, and the following section was added to the Discussions:

“Bile acids (BAs) metabolism has been associated with gut microbiota composition in many studies. Indeed, the gut microbiota shapes the composition of the BA pool (by hydrolysis and hydroxy group dehydrogenation of primary BAs to secondary BAs) and BAs can affect the growth of certain gut bacteria³⁵⁻³⁷. In faeces, 5% and 3% of the total number of associations between faecal metabolites and metabolic pathways and species, respectively, were with BAs, over 80% of which were with secondary BAs. In blood, 6% of all associations with species and 3% of all associations with metabolic pathways were with BAs. Again, secondary BAs were more associated (over 70% of all BAs associations) than primary BAs with both species and metabolic pathways.”

Another point with relevance to the interpretation of the data is in the discussion “90% of the microbial species interact with their surrounding metabolic environment“ - I fail to see how the other 10% survive.

Answer: This was indeed an unfortunate wording. We now write in the Discussions Section:
“At 5% FDR, we identified association between the faecal metabolites and 90% of the microbial species and 99.7% of the microbial metabolic pathways.”

In the abstract, it would be helpful to mention which similarity measure is being applied.

Answer: Similarity was evaluated as the ratio between the number of species/pathways present in both members of the unrelated pairs of individuals, and the number of species/metabolic pathways that were present in at least one of the members of the pair. Following the Reviewer's suggestion, we now write in the abstract:

“We observed that a random pair of unrelated subjects shared, on average, a far greater number of functional metabolic pathways (82%) than species (43%).”

and in the Methods Section:

“Shared species and microbial metabolic pathways between unrelated individuals. For all individuals in our datasets, we codified the absence/presence of a microbial species/metabolic pathways with 0 and 1, respectively. Then, after having identified all possible pairs of unrelated individuals (N=1,006,288), we assessed for each pair the percentage of shared species/pathways as the ratio between the number of species/pathways which were present in both members and the number of species/metabolic pathways which were present in

at least one of them. The distribution of percentages obtained for species and pathways across all pairs were then compared using a paired Wilcoxon's test."

Similarly, it would be helpful to mention the method employed for association testing in the results section and the legend of figure 2. It may be obvious to researchers from the human genetics field, but after reading the presentation of results from unrelated subjects in the first part of the results section, I didn't realize that the association analysis actually uses kinship information. On a related note, the methods section should state whether pedigree information or genomic data was used in the association models.

Answer: We agree with the Reviewer and edited the Results Section as:

*"713 annotated metabolites were measured in more than 50 individuals and tested for association with the gut microbiome at both taxonomic and functional levels using PopPAnTe¹², which uses a variance component framework and the matrix of the expected kinship between each pair of individuals to model the resemblance between family members. Sex and age at sample collection were included as covariates (**Methods, Figure 2**)."*

We also added the following sentences to the legend of Figure 2:

"Association testing was performed using PopPAnTe¹², in order to model the resemblance between family members. Sex and age at the sample collection were included as covariates."

and in the Methods Section:

"Associations of faecal and blood metabolites with species and microbial metabolic pathways transformed relative abundances were carried out using PopPAnTe¹², which uses a variance component framework and the matrix of the expected kinship between each pair of individuals, generated using the pedigree information, to model the resemblance between family members."

The methods section is lacking details on sampling, and especially storage conditions and duration. It is also unclear whether faecal samples for metagenomics and metabolomics are the same samples or were taken at different points in time. Similarly, it is unclear whether the blood samples were taken at the same time.

Answer: We acknowledge that these information were overlooked in the previous version of the manuscript. Twins collected their fecal samples at home, and the samples were refrigerated for up to 2 days prior to their annual clinical visit at King's College London, at which point they were stored at -80°C for an average of 2.3 ± 1.0 years at -80°C before processing. Blood samples were stored at -80°C for an average of 1.8 ± 1.2 years before processing. The faecal samples used for metagenomics profiling were also used for metabolomics profiling, while blood metabolites were profiled using samples collected on average 0.9 ± 1.3 years apart. We now write, in the Methods Section:

"The TwinsUK adult twin registry includes about 14,000 subjects, predominantly females, with disease and lifestyle characteristic similar to the general UK population⁵¹. Metagenomics sequencing was performed on 1,054 randomly selected samples, while faecal and blood

metabolomics was assessed in 479 and 859 individuals with metagenomics data, respectively. Twins collected fecal samples at home, and the samples were refrigerated for up to 2 days prior to their annual clinical visit at King's College London, when they were stored at -80°C for an average of 2.3 ± 1.0 at -80°C before processing. Both faecal metabolomics and WMGS data were generated on the same faecal samples. Blood samples, collected during the clinical visit, were stored at -80°C for an average of 1.8 ± 1.2 years before processing. Faecal and blood samples were collected, in average, 0.9 ± 1.3 years apart."

Numbers of reads: an average of 39 M QC'ed and filtered reads is mentioned in the abstract and the results section. However, in the methods section, it is mentioned that roughly 27M reads were sequenced per samples were sequenced. These numbers don't add up.

Answer: We agree with the Reviewer that the reporting of the average number of reads was confusing. 27M is the average number of paired-end reads sequenced (54M reads in total), while 39M refers to the average number of reads after quality control, which discarded problematic samples (including four samples with less than 15M reads), and which does not distinguish anymore between forward and reverse reads. We now write:

"Sequencing of 1,054 samples yielded an average number of reads of 54M per sample before quality control."

and maintained that:

"Finally, we removed individuals not of European ancestry (N=9, self-reported via questionnaire) resulting in 1,004 samples with an average number of reads of 39M."

Figure 2: the N stated at the top of the figure is misleading. Indeed, none of the displayed analyses were performed with 1004 individuals, as the original cohort was larger and the cohorts used for the two analyses are both considerably smaller. The 1004 should be replaced with the sample numbers for both arms.

Answer: We modified Figure 2 following the Reviewer's comment. We removed N=1004 and included "N = 479" on the faecal metabolome side, and "N = 859" on the blood metabolome side, as reported below:

Caption: Study design and number of associations between the gut microbiome and the faecal and blood metabolome. The top of the Figure reports the number of microbial species and metabolic pathways which were detected in at least 50 individuals with metabolomics and WMGS data, and that were used in the study, and the number of associations tested. The bottom of the Figure reports the number of associations that were significant at an FDR 5%, along with the number and percentage of metabolites, microbial species, and microbial metabolic pathways involved. Association testing was performed using PopPAnTe¹², in order to model the resemblance between family members. Sex and age at the sample collection were included as covariates.

On a related note, for the comparisons of associations in blood and faecal samples, were all results used or only those relating to the cohort that had faecal and blood metabolites measured?

Answer: To fully exploit these datasets, we carried out the metagenome-wide association studies using all the available samples for both faeces and blood. Conversely, the P-gain statistics was assessed using subjects having both faecal and blood metabolomic profiles available.

We now clarify this in the Methods Section (Metagenome-wide association study):

“Only pairs of metabolites-species/pathways with at least 50 observations were tested for association. In these analyses, we used all the available samples with faecal metabolites (N=479) and with blood metabolites (N=859).”

and (Gut-host metabolic dialogue)

“All association tests were carried out between pairs of co-associated metabolites and metagenomic data with at least 100 complete observations (i.e., having metagenomic data and metabolic profile for both co-associated metabolites available).”

The differences of correlations in the absence or presence of species could be due to common reasons for correlation and species presence. Eg. if metabolite comes from a food source that also allows for the microbial species' growth. This test for differences in correlation therefore is no proof of interaction.

In light of the Reviewer's comment, we realised that the description of this analysis step in our manuscript has been kept too short, and it is consequently slightly unclear. After identifying multiple associations between the microbiome and both faecal and blood metabolites, we wondered whether associations between the same microbiota (or pathway) and metabolites in both blood and faeces were enriched for independent blood and faecal associations (particularly given that the blood and faecal metabolites were often not the same), or whether they were actually suggesting that the faecal metabolite - blood metabolite - species/pathway trios were in some way interconnected. Therefore, with this test we were not seeking to test for interaction, but simply to justify and support our choice of performing a P-gain analysis focussed on these associated trios. The differential correlation analysis suggested that at least some of the observed associations between a species (or pathway) and metabolites in both faeces and blood were likely not independent signals, and we felt justified to proceed with the P-gain analysis using these associated trios.

To make it more clear we have now modified the Results Section:

"These results suggested that at least some of the observed faecal and blood metabolite associations were likely not randomly coincident at the same species (or pathway), thus supporting the analysis of this subset of faecal metabolite - blood metabolite - species/pathway trios with the P-gain approach."

Figure 3: the arrows for the bile acids are a bit confusing. Bile acids aren't produced in the blood, as is suggested, and they are transformed by the microbiome, so the arrows shouldn't end at the microbiome.

Answer: We modified Figure 3 following the Reviewer's comment.

Caption: Summary of possible mechanisms of host-microbiome metabolic dialogue. The Figure highlights the possible four mechanisms implicated in the interplay between the gut microbiome, the faecal metabolome, and the blood metabolome. (1) Small dashed lines: metabolites are produced by the microbiota and then absorbed, resulting in associations between the microbiome and both the blood and faecal metabolites. (2) Large dashed lines: the microbiome affects the gut barrier integrity, resulting in alteration of metabolites absorption (i.e., the same metabolite is associated with a species/pathway in both blood and faeces, but the directions of effects are opposite). (3) Light continuous line: metabolites produced by the host, such as bile acids, affect microbial growth. (4) Bold continuous line: direct microbiome to host cell interactions that results in host systemic modulation (i.e., species are associated with blood metabolites but not with faecal metabolites).

The discussion lists some limitations of the presented study. I would argue that another limitation is that the microbiome analyses are not quantitative, i.e. there is no estimate of microbial density and therefore it is impossible to say if a relative abundance of one pathway in one sample is more or less than the abundance in another sample.

Answer: We agree with the Reviewer that using relative abundances instead of actual concentrations is another limitation of the study and that this should be mentioned in the Discussions. We have now added the following sentence to the Discussions:

“Finally, the results obtained in this study are not quantitative, since all analyses were carried out using relative abundances. This imply that the identified associations report the effect of microbial species/metabolic pathways proportion rather than of their actual concentration.”

**Language:
throughout:**

- *I would replace “gut lumen metabolic content” (and similar) by “faecal metabolic content”, because that is where the metabolites were measured. Also, gut lumen could refer to different gut sections and is therefore less precise.*
- *there are several instances where the authors refer to pathways or species “affecting” or “controlling” metabolite abundances. Since the analyses are associative, these terms are over-interpretative and should (at least in the results section) be replaced by more appropriate choices.*

Answer: We agree with this Reviewer. We have now changed all occurrences of “*gut lumen metabolic content*” with “*faecal metabolic content*” or similar expressions. Also, we have substituted the words “*affecting*” and “*controlling*” with “*associating*”.

Intro, paragraph 1:

- *“variations in its composition induces” should be “induce”*

Intro, paragraph 2:

- *“16S amplicon data” should be “16S rRNA gene amplicon data” (or “16S rRNA amplicon data”, if it refers to RNA-based assays)*
- *“host’s health” should be “hosts’ health” or “host health”*

Gut microbiota composition is host-specific whereas its functions are shared across subjects:

par 1:

- *should be “Subdoligranulum genus”*
- *“Microbial metabolic profiling” is a bit misleading. “Microbial metabolic pathway detection” might be more to the point.*

The microbiome is involved in the crosstalk between the gut lumen and host systemic metabolism:

par. 1, last sentence - introduce singular article for “species” or change “was” to “were”

par. 2, 3rd sentence - “two metabolite levels” would be more clear than “two metabolites”, since this can refer to the same metabolite in two matrices.

Answer: We are grateful to the Reviewer for the the thoughtful review. We have now fixed these mistakes.

Gut microbiota composition is host-specific whereas its functions are shared across subjects:

par. 2:

- “pathway prevalence within our sample strongly correlated with the number of species in which it could be detected” - does the detection refer to the same sample or the overall dataset?

Answer: The detection refers to the whole dataset. To improve clarity, we amended the sentence as follow:

“As a consequence, pathway prevalence strongly correlated with the number of species in which it could be detected in the overall dataset (Spearman’s $\rho = 0.34$; $P=9.4 \times 10^{-9}$), i.e., pathways present in the largest number of species were also those with the highest prevalence (and vice versa)”.

The microbiome is involved in the crosstalk between the gut lumen and host systemic metabolism:

- *In my opinion, the term “crosstalk” is inadequate here, as it indicates some kind of signalling process. It would probably help to define already here what kind of mechanisms could lead to the observed patterns and choose a term that reflects them.*

Answer: We agree that the term “crosstalk” may be misleading, and we now use “dialogue” or “communication”.

Discussion - end of par. 3 - are the associations with vitamin B metabolites in blood or faeces or both?

Answer: These associations were with the faecal metabolites. We now write: *“B vitamins in faeces were strongly associated with both species and metabolic pathways, with riboflavin (vitamin B2), nicotinate (vitamin B3), pantothenate (vitamin B5), pyridoxine (vitamin B6), biotin (vitamin B7) associated with 9 to 27 species and with 48 to 155 microbial pathways (Supplementary Data D2 and D3). Finally, 16 associations were observed between faecal vitamin E (alpha, beta, gamma and delta tocopherol) and species/pathways.”*

Reviewers' comments:

Reviewer #1 (Remarks to the Author):

In the revised manuscript the authors have satisfactorily addressed the technical issues I had raised in my original review. However, still the manuscript would be markedly improved by specifying the analytical platforms where each metabolite was identified. The authors might consider to create one more column clarifying the type of platform applied along with each metabolite in the supplementary data, which is necessary for other investigators to choose technical approach when performing similar profilings.

Reviewer #2 (Remarks to the Author):

The revised manuscript by Visconti et al. is much improved and I am satisfied by the responses to my earlier questions. I thank the authors for their careful revisions.

However, I am concerned about the sampling design. It is now stated that the faecal and blood samples were taken on average 0.9 +/- 1.3 years apart. Given the variation of at least some proportion of blood metabolites (e.g. doi 10.1016/j.molmet.2018.06.008), and also the microbiome (e.g. 10.1128/mSystems.00144-16), I would expect that many of the reported associations are false positives.

Do the authors have any estimates on the variability of the microbial taxa and the metabolites which they find associated from which they could estimate effect sizes that are robust to temporal variability?

Reviewers' comments:

Reviewer #1

In the revised manuscript the authors have satisfactorily addressed the technical issues I had raised in my original review. However, still the manuscript would be markedly improved by specifying the analytical platforms where each metabolite was identified. The authors might consider to create one more column clarifying the type of platform applied along with each metabolite in the supplementary data, which is necessary for other investigators to choose technical approach when performing similar profilings.

Answer: We are glad the Reviewer was satisfied with our previous revisions. We agree that to specify the analytical platform used for each metabolite would help reproducing the present findings. Therefore, we included two supplementary data files (Supplementary Data D19 and D20; for the faecal and blood metabolites, respectively), which specify, for each metabolite, the type of platform used.

We have now added the following sentence to the Methods Section:

“Details regarding the platform used for each individual metabolite are provided as Supplementary Data D19 and D20.”

Reviewer #2

The revised manuscript by Visconti et al. is much improved and I am satisfied by the responses to my earlier questions. I thank the authors for their careful revisions.

However, I am concerned about the sampling design. It is now stated that the faecal and blood samples were taken on average 0.9 +/- 1.3 years apart. Given the variation of at least some proportion of blood metabolites (e.g. doi 10.1016/j.molmet.2018.06.008), and also the microbiome (e.g. 10.1128/mSystems.00144-16), I would expect that many of the reported associations are false positives.

Do the authors have any estimates on the variability of the microbial taxa and the metabolites which they find associated from which they could estimate effect sizes that are robust to temporal variability?

Answer: We are glad the Reviewer was satisfied with our previous revisions. We agree with the Reviewer's concern regarding the temporal stability of the metagenomic and metabolomic data. We did not collect longitudinal faecal metagenomics data, therefore making it impossible to assess metagenomic stability in our study. However, this has already been shown to remain relatively stable during adulthood [DOI: 10.1073/pnas.1423854112, DOI: 10.1126/science.1237439] in the absence of external perturbations.

Regarding the blood metabolomic data, while these and the metagenomic data were taken on average 0.9 years apart, the median difference was 0.43 years, with 38% of our samples were taken within 3 days, 41% within one week, and, in general, 68% of our measurements were taken at no more than one year apart and 91% of them at no more than two years apart, as now shown on Supplementary Figure 4 (also reported below).

We assessed metabolomic stability over time using longitudinal measurements (up to three time points) of the blood metabolomics used in this study, which were available for a larger set of 2,070 individuals. As already reported in our manuscript (and in [DOI: 10.1038/ng.3809]), measurements were carried out by Metabolon.

Since over 90% of metabolomics and metagenomics data were measured no more than 2 years apart, we extracted all the individuals having two measurements within a 2-year time frame (N=149), ensuring that their metabolomic profiles were assessed in the same batch, in order to limit potential variability due to batch effects. We then removed, for each tested metabolite's profile, outliers (values further away than 3 standard deviations from the dataset mean), scaled the data to have mean zero and standard deviation one, and assessed the intra-individual correlations. We observed an average intra-individual correlation of 0.53 (median: 0.55, SD: 0.12, 1st-3rd interquartile range: 0.47-0.60). When limiting the analysis to the blood metabolites which were significantly associated either to microbiome species or functions (instead of using the whole set of metabolites that has been tested in our study), we obtained very similar results (average intra-individual correlation: 0.53, median: 0.54, SD: 0.13, 1st-3rd interquartile range: 0.47-0.62).

To further verify the stability of the intra-individual correlation over larger time frames, we repeated the same experiment using increasing number of years between the first and the second metabolomics measurement, that is at 3, 4, 5, ..., and up to 10 years apart. The number of pairs used within each time frame (N), and the average intra-individual correlations, along with their

standard error, SD, and 1st-3rd interquartile (IQ) range, are now reported in Supplementary Table 2 (also below). We observed stability of the metabolomic profiles over larger time frames, in line with the results from another study which showed that the human metabolic profile is conserved for up to seven years [DOI: 10.1007/s11306-014-0629-y].

Finally, to verify that the observed correlation were not due to chance, we compared the average intra-individual correlation observed in the 149 individuals with measurements taken 2 years apart with that observed in 149 randomly paired measurements from unrelated subjects extracted from the whole metabolomics dataset (and measured in the same batch). Then, we used the Wilcoxon's test to assess the probability of observing a greater correlation between the metabolic profiles of the same individual at different time points compared to the random pairs. We obtained an empirical $P=1 \times 10^{-4}$, with all the random datasets showing negligible correlation (mean $\rho=0.03$; max $\rho=0.09$).

We now write in the Results Section

“Faecal and blood samples were collected, on average, 0.9 years apart, with 41% of our samples collected within one week, and 91% within two years (Supplementary Figure 4). Intra-individual correlation analysis of tested metabolites showed a good correlation between samples collected up to 2 years apart (N=149, mean Pearson's $\rho=0.53$, SD: 0.12, 1st-3rd interquartile range: 0.47-0.60), as confirmed by a permutation analysis ($P_{\text{empirical}}=1 \times 10^{-4}$, Methods). We further observed that metabolomics stability persists over longer periods of time (Supplementary Table 2), in line with previous literature suggesting that human metabolic profiles are conserved for up to seven years¹³.”

and in Methods Section:

“Temporal stability of the metabolic profiles

The blood metabolomic data used in this study belonged to a larger set of 2,070 individuals, with longitudinal measurements up to three time points³³, which we used to assess the blood metabolomic stability over time.

In line with the difference observed between the metagenomic and blood metabolomic data used in this study, where about 90% of our samples were collected no more than 2 years apart (Supplementary Figure 4), we extracted all the individuals having two measurements within a 2-year time frame (N=149), and ensuring that their metabolomic profiles were assessed in the same batch in order to limit potential variability due to batch effects. We then removed, for each tested metabolite's profile, outliers (values further away than 3 standard deviations from the dataset mean), scaled the data to have mean zero and standard deviation one, and assessed the intra-individual correlations using the Pearson's ρ . To confirm that the observed correlations were not due to chance, we then built 10,000 datasets including 149 randomly paired metabolomic profiles from unrelated subjects extracted from the whole metabolomics dataset, ensuring that each pair was measured in the same batch. We then used the Wilcoxon's test to assess the probability of observing a greater average intra-individual correlation in the 149 individuals with measurements taken 2 years apart compared to that observed the random sets. To further verify the stability of the intra-individual correlation over larger time frames, we evaluated the intra-individual correlation for measurements taken up to 10 years apart (and within the same batch).”

Caption: Distribution of time point difference between metagenomic and blood metabolomic data. Mean difference was 0.9 years (median: 0.43, SD: 1.3).

Caption: Intra-individuals correlation of blood metabolomics profiles over time. The table reports, for measurements taken up to 10 years apart, the number of individuals used within each time frame (N), and the mean intra-individual correlations, along with their standard error (SD), and 1st-3rd interquartile (IQ) range.

Years apart	N	Mean	SD	1st-3rd IQ range
2	149	0.53	0.12	0.47-0.60
3	180	0.52	0.13	0.43-0.62
4	282	0.52	0.13	0.43-0.61
5	352	0.50	0.13	0.42-0.60
6	446	0.48	0.13	0.40-0.57
7	552	0.49	0.12	0.41-0.58
8	506	0.49	0.13	0.41-0.59
9	331	0.48	0.13	0.40-0.58
10	139	0.48	0.12	0.39-0.58

REVIEWERS' COMMENTS:

Reviewer #2 (Remarks to the Author):

I think the problem of gut and blood samples not having been taken at the same time is now transparently discussed. The measures taken by the authors to validate their results in light of this problem are appropriate.

Reviewers' comments:

Reviewer #2

I think the problem of gut and blood samples not having been taken at the same time is now transparently discussed. The measures taken by the authors to validate their results in light of this problem are appropriate.

Answer: We are glad the Reviewer was satisfied with our revisions.